# Delivery of ceramide phosphoethanolamine lipids to the cleavage furrow through the endocytic pathway is essential for male meiotic cytokinesis

Govind Kunduri[1]*, Si-Hung Le[2], Valentina Baena[3,4], Nagampalli Vijaykrishna[5], Adam Harned[3,4], Kunio Nagashima[3,4], Daniel Blankenberg[5], Izumi Yoshihiro[2], Kedar Narayan[3,4], Takeshi Bamba[2], Usha Acharya[1]*, Jairaj K. Acharya[1]*

1 Cancer and Developmental Biology Laboratory, National Cancer Institute, Frederick, Maryland, United States of America, 2 Division of Metabolomics, Medical Institute of Bioregulation, Kyushu University, Fukuoka, Japan, 3 Center for Molecular Microscopy, Center for Cancer Research, National Cancer Institute, National Institutes of Health, Bethesda, Maryland, United States of America, 4 Cancer Research Technology Program, Frederick National Laboratory for Cancer Research, Frederick, Maryland, United States of America, 5 Genomic Medicine Institute and Lerner Research Institute, Cleveland Clinic, Cleveland, Ohio, United States of America

* kunduri.govind@nih.gov (GK); usha.acharya@nih.gov (UA); acharyaj@mail.nih.gov (JKA)

**Data Availability Statement:** The RNA-sequencing data has been deposited at the GEO (GSE208655).

## Abstract

Cell division, wherein 1 cell divides into 2 daughter cells, is fundamental to all living organisms. Cytokinesis, the final step in cell division, begins with the formation of an actomyosin contractile ring, positioned midway between the segregated chromosomes. Constriction of the ring with concomitant membrane deposition in a specified spatiotemporal manner generates a cleavage furrow that physically separates the cytoplasm. Unique lipids with specific biophysical properties have been shown to localize to intercellular bridges (also called midbody) connecting the 2 dividing cells; however, their biological roles and delivery mechanisms remain largely unknown. In this study, we show that ceramide phosphoethanolamine (CPE), the structural analog of sphingomyelin, has unique acyl chain anchors in *Drosophila* spermatocytes and is essential for meiotic cytokinesis. The head group of CPE is also important for spermatogenesis. We find that aberrant central spindle and contractile ring behavior but not mislocalization of phosphatidylinositol phosphates (PIPs) at the plasma membrane is responsible for the male meiotic cytokinesis defect in CPE-deficient animals. Further, we demonstrate the enrichment of CPE in multivesicular bodies marked by Rab7, which in turn localize to cleavage furrow. Volume electron microscopy analysis using correlative light and focused ion beam scanning electron microscopy shows that CPE-enriched Rab7 positive endosomes are juxtaposed on contractile ring material. Correlative light and transmission electron microscopy reveal Rab7 positive endosomes as a multivesicular body-like organelle that releases its intraluminal vesicles in the vicinity of ingressing furrows. Genetic ablation of Rab7 or Rab35 or expression of dominant negative Rab11 results in significant meiotic cytokinesis defects. Further, we show that Rab11 function is required for localization of

All other relevant data are within the paper and its supporting information files.

**Funding:** This study was funded by the intramural division of the National Cancer Institute, National Institutes of Health (Division of Health and Human Services) to J.K.A. The content of this publication does not necessarily reflect the views or policies of the Department of Health and Human Services, nor does mention of trade names, commercial products, or organizations imply endorsement by the U.S. Government. The study was partly funded with Frederal funds from the National Cancer Institute, National Institutes of Health, under Contract No. 75N91019D00024 (to K.N.). The work was partially supported by the Grant-in-Aid for Scientific Research on Innovative Areas (17H06304) [T.B.] and a Grant-in-Aid for Scientific Research (B) (18H01800) [T.B.] from Japan Society for the Promotion of Science (JSPS). The funders had no role in study design, data collection and analysis, decision to publish, or preparation of the manuscript.

**Competing interests:** The authors have declared that no competing interests exist.

**Abbreviations:** CAPT, CDP-alcohol phosphotransferase; CLEM, correlative light-electron microscopy; CERT, ceramide transfer protein; CPE, ceramide phosphoethanolamine; CPES, ceramide phosphoethanolamine synthase; DHSPH, dihydrosphingosine; ER, endoplasmic reticulum; GSC, germline stem cell; GSEA, Gene Set Enrichment Analysis; IC, individualization complex; MUFA, monounsaturated fatty acid; MVB, multivesicular body; MVE, multivesicular endosome; PA, phosphatidic acid; PE, phosphatidylethanolamine; PIP, phosphatidylinositol phosphate; PS, phosphatidylserine; RLC, regulatory light chain; ROI, regions of interest; SD, standard deviation; SFA, saturated fatty acid; SM, sphingomyelin; SPD, sphingadiene; SPH, sphingosine; SPT, serine palmitoyltransferase; UTR, untranslated region; VLC, very long chain; VLC-PUFA, very long chain polyunsaturated fatty acid.

CPE positive endosomes to the cleavage furrow. Our results imply that endosomal delivery of CPE to ingressing membranes is crucial for meiotic cytokinesis.

## Introduction

Eukaryotic cells display high diversity in lipid species that are chemically and structurally distinct. They are classified into 8 major categories, each of which is further subdivided into classes and subclasses [1]. Lipids usually comprise of a polar head group that is linked by a structural backbone to a hydrophobic tail composed of acyl chains. Cellular lipid diversity arises from the large number of combinatorial chemical possibilities that the head and tail moieties can generate. Diversification of lipids to cope with evolving cellular complexity has resulted in thousands of lipid species with purported unique functions, most of which still remain unknown [2].

The head groups of each lipid class have been widely recognized for their biological function. For example, the inositol head group of phosphatidylinositol phosphates (PIPs) and its modifications play critical roles in cellular signaling [3]. However, even within each lipid class, there are molecularly distinct species that have identical head group but differ extensively in their acyl side chains, such as number of carbon atoms, number of double bonds, and the nature of the chemical linkage (e.g., ester or ether). An increasing number of studies are beginning to reveal the importance of acyl side chains by demonstrating that they selectively bind to target proteins and elicit distinct biological functions [4,5]. For example, a recent study showed that the acyl side chains of diacylglycerol significantly influence the recruitment of protein kinase C to the plasma membrane [6]. Similarly, the chain length of ceramide was shown to be critical for selective protein cargo sorting at endoplasmic reticulum (ER) exit sites in yeast [7]. The COPI machinery protein p24 bound specifically to N-stearoyl sphingomyelin (SM C18) and functioned as a cofactor to regulate COPI-dependent transport in the early secretory pathway. Further, this interaction depended on both the head group and the backbone of the sphingolipid [8].

Cytokinesis is the final step in cell division that divides 1 cell into 2 daughter cells. During cytokinesis, an actomyosin contractile ring is formed that is positioned midway between the segregated chromosomes. Narrowing of this ring with the addition of membrane in a spatio-temporally defined mode produces a cleavage furrow that physically divides the cytoplasm [9]. During somatic cytokinesis, the 2 daughter cells are interconnected via intercellular bridges prior to abscission. Although these structures are transient, they accumulate distinct lipid species to mediate correct cell division [10–12]. Unlike somatic cytokinesis, in testis, the developing spermatogonial cells divide synchronously with an incomplete cytokinesis where all the daughter cells remain interconnected by cytoplasmic bridges. Thus, the spermatogonial cells develop as a syncytium and become separated from each other only at the end of spermatogenesis during spermatid individualization [13]. Lipids have been shown to participate in cytokinetic furrow ingression, midbody structure stabilization, and abscission during cytokinesis [14,15]. The high degree of membrane curvature necessary for cleavage furrow ingression and stabilization of intercellular bridges (ICB)/cytoplasmic bridges necessitates lipid components with specific biophysical properties. While the importance of very long chain fatty acids (which are components of sphingolipids and glycosphingolipids) in cytokinesis has been shown, the relationship between membrane lipid composition, ring constriction, and furrow ingression is still underexplored [14,16]. Although sphingomyelins containing very long chain

polyunsaturated fatty acids (VLC-PUFAs) have been identified in various mammalian testes including humans, their direct role in spermatogenesis, particularly in meiotic cytokinesis remains unknown [17,18]. Other lipids including PIPs, cholesterol, and phosphatidylethanolamine (PE) have been shown to be enriched at the cytokinetic furrow [14]. Midbodies also accumulate specific lipids including sphingolipids, phosphatidylserine (PS), phosphatidic acid (PA), and even unique triacylglycerols [12,15]. However, specific mechanisms involved in the delivery of these lipids to the cytokinetic furrow or midbodies remain poorly understood. Membrane traffic via exocytic and endocytic mechanisms has been shown to be essential for successful completion of cytokinesis [19–23]. However, only few studies have focused on individual lipid trafficking via exocytic pathways [24–26] and whether endocytic pathways play direct roles in delivering specific lipids to the cytokinetic furrow or midbody remain unknown.

Ceramide phosphoethanolamine (CPE) is a structural analog of sphingomyelin (SM) in *Drosophila melanogaster*. CPE is synthesized in the luminal compartment of trans Golgi by ceramide phosphoethanolamine synthase (CPES) (Fig 1A). Previously, we have shown that *cpes* null mutants show significant late pupal lethality during development and only about 25% of mutants survive to adulthood. About 60% of *cpes* mutant adults have dorsal closure defects and all the mutant males are sterile. Aged *cpes* mutant adults display light inducible seizures and paralysis due to defective cortex glia [27]. In this study, we show that CPE has unique acyl chain anchors in spermatocytes and is essential for meiotic cytokinesis, spermatid polarity, and individualization. The head group of CPE is also important for spermatogenesis. CPES expression in transit amplifying to the spermatocyte stage is essential for successful completion of spermatogenesis. We show that aberrant central spindle behavior, but not mislocalization of PIPs at the cleavage furrows, is responsible for the meiotic cytokinesis defect in *cpes* mutants. Further, we show that endocytically retrieved CPE from the plasma membrane is enriched in Rab7 positive multivesicular bodies that dock to the ingressing membranes and release intraluminal vesicles in their vicinity. Our results demonstrate the importance of CPE-rich membrane addition at the cleavage furrow involving the endocytic pathway.

## Results

### Sphingosine with VLC-mono-unsaturated fatty acid and sphingadiene with VLC-saturated fatty acid containing sphingolipids are enriched in the testis

To investigate sphingolipid species diversity and their role in *Drosophila* tissues, we performed supercritical fluid chromatography coupled to mass spectrometry (SFC/MS/MS) on lipids from wild-type fly head, dissected ovary, and testis (S1 Data). As shown in Fig 1B, CPE is a more abundant sphingolipid compared to ceramides (Fig 1C) and hexosylceramides (HexCer) (Fig 1D). The amount of CPE was significantly higher in the testis compared to the head and ovary (Fig 1B). In contrast, hexosylceramides were significantly higher in heads compared to ovary and testis (Fig 1D).

Closer examination of acyl chain composition revealed that head and ovary-derived CPEs are predominantly composed of sphingosine linked to saturated fatty acid (SPH_SFA) (Fig 1E and 1F). Interestingly, in addition to SPH_SFA, CPE in the testis is enriched in 2 distinct species with net 2 double bonds in their acyl chain composition including sphingosine (1 double bond) linked to monounsaturated fatty acid (SPH_MUFA) (Fig 1E and 1G) and a sphingadiene (2 double bonds) linked to saturated fatty acid (SPD_SFA) (Fig 1E and 1H). The fatty acid acyl chain lengths of these CPE species are predominantly C20, C22, C24, and therefore could be classified as very long chain (VLC) sphingolipids. The sphingoid base acyl chain length is d14 and d16 (Fig 1G and 1H). The corresponding ceramide precursors are also

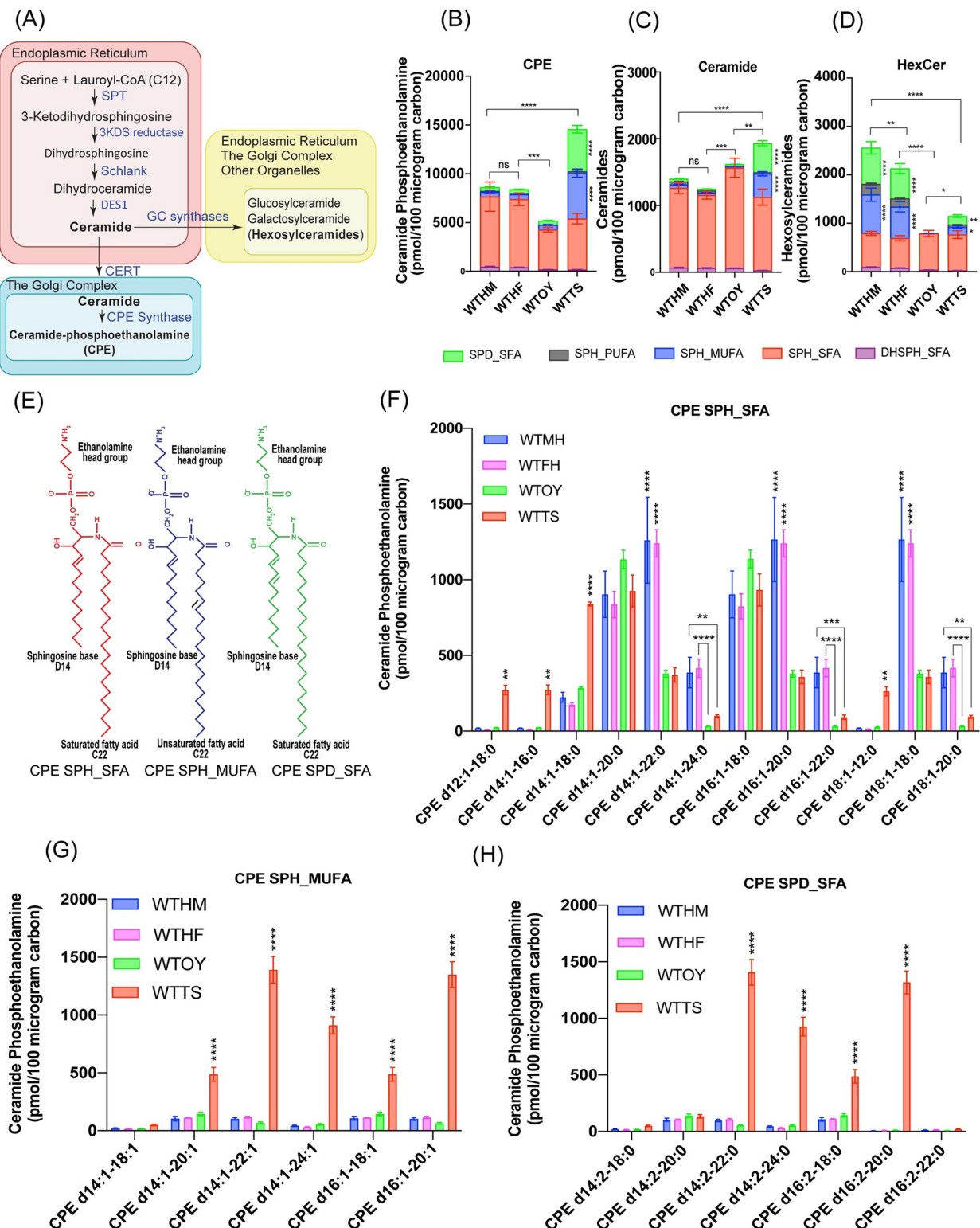

**Fig 1. Sphingolipid analysis in wild-type *Drosophila* tissues.** (A) De novo sphingolipid biosynthetic pathway. De novo sphingolipid biosynthetic pathway begins with condensation of an amino acid serine and a fatty acid, lauryl-CoA into 3KDS by SPT complex in the membranes of ER. 3KDS is reduced to dihydrosphingosine by 3KDS reductase. Dihydrosphingosine is acylated to produce dihydroceramide by Schlank. Dihydroceramide is desaturated by Δ4 desaturase (DES1) resulting in the formation of ceramide. Ceramide acts as a precursor for the biosynthesis of complex sphingolipids both in the ER and Golgi complex. Ceramide is actively transported to Golgi by CERT. In the trans Golgi lumen, ceramide is

converted to hexosylceramides by GC synthases and to CPE by CPE synthase. (B-D) Sphingolipids were extracted from wild type male and female heads and dissected ovaries and testes using Methanol:Chloroform method and subjected to SFC/MS/MS. WTHM, WTHF, WTOY, and WTTS. Sphingolipids were quantitated by measuring large fragments in MRM, and average amounts from 3 independent biological replicates are shown as picomoles (pmol) in 100 micrograms of carbon. Each sphingolipid species (CPE (1B), Ceramide (1C) and HexCer (1D)) was further subdivided into 5 major subspecies. Each of the subspecies differed in acyl chain length but similar in chemical properties: DHSPH base linked to SFA (DHSPH_SFA), SPH base linked to SFA (SPH_SFA), SPH base linked to MUFA (SPH_MUFA), SPH base linked to PUFA (2–6 double bonds) (SPH_PUFA), and SPD with SFA (SPD_SFA). (E) Cartoon showing the chemical structure of 3 major CPE subspecies including CPE_SPH_SFA, CPE_SPH_MUFA, and CPE_SPD_SFA. (F) CPE subspecies within CPE_SPH_SFA that differed in number of carbon atoms in acyl chains of sphingosine base and fatty acid. (G) CPE subspecies within CPE_SPH_MUFA. (H) CPE subspecies within CPE_SPD_SFA. Statistical significance was calculated using mean, SD, and N in Prism 8. The 2-way ANOVA multiple comparison was used to calculate $p$-values where **** $p \leq 0.0001$, *** $p \leq 0.001$, ** $p \leq 0.01$, * $p \leq 0.05$, and ns $p > 0.05$. Three independent biological replicates were performed for each sample and lipids were extracted from 400 heads or 250 pairs of ovaries or testes for each biological replicate. The data underlying the graphs shown in this figure can be found in S1 Data. 3KDS, 3-ketodihydrosphinganine; CERT, ceramide transfer protein; CPE, ceramide phosphoethanolamine; DHSPH, dihydrosphingosine; ER, endoplasmic reticulum; MUFA, monounsaturated fatty acid; PUFA, polyunsaturated fatty acid; SD, standard deviation; SFA, saturated fatty acid; SPD, sphingadiene; SPH, sphingosine; SPT, serine palmitoyltransferase; WTHF, wild-type heads from female; WTHM, wild-type heads from male; WTOY, wild-type ovary; WTTS, wild-type testis.

evident in the wild-type testis samples indicating acyl chain diversity is likely introduced during ceramide synthesis but not after CPE synthesis (Fig 1C and S1 Data). Remarkably, SPH_MUFA and SPD_SFA are also present in HexCer but at significantly lower levels in the testis compared to heads (Fig 1D versus Fig 1B). The fatty acid acyl chain length of HexCer_SPH_MUFA and HexCer_SPD_SFA showed that they are predominantly C18 and C20 while the sphingoid base is primarily d14 and d16 (S1B, S1D, and S1E Fig). HexCer with d14 sphingosine base and C22 PUFA (SPH-PUFA) is enriched in heads compared to testis (Figs 1D, S1C, and S1E). Taken together, these results suggest that CPE in the testis and HexCer in the central nervous system have a unique acyl chain composition.

## CPE-deficient males are sterile and show defects in male meiotic cytokinesis, spermatid polarity, and individualization

To understand the biological significance of the unique CPE species in the testis, we reasoned that the *cpes* null mutant generated by us would serve as a suitable tool since *cpes* mutant males are sterile [27]. Immunostaining of *cpes* mutant testis with antibody to Vasa protein, a germ cell-specific conserved RNA helicase [28], showed accumulation of germ cells at the tip of the testis (Fig 2B) and DNA-specific staining with DAPI showed absence of mature sperm in seminal vesicles. To investigate cytological defects in *cpes* mutant testis, we performed live testis squash preparations followed by phase contrast microscopy. This method allows for easy identification of almost all of the major stages of spermatogenesis [29]. During meiotic divisions, chromosomes and mitochondria are equally partitioned to each of the 4 daughter cells. Immediately following meiosis, in the round spermatid stage, mitochondria in each daughter cell aggregate to form a large phase dark structure known as the nebenkern. Round spermatids (onion stage spermatids) are characterized by phase dark nebenkern and phase light nucleus at roughly equal size and at a ratio of 1:1 per daughter cell. The nuclear size of round spermatids is directly proportional to the chromatin content; hence, any errors in chromosome segregation during meiosis can lead to variability in nuclear size [30,31]. Similarly, defects in meiotic cytokinesis following normal chromosome segregation results in cells with 2 or 4 nuclei and an abnormally large nebenkern. This change can be easily visualized by phase contrast microscopy and is often used as a diagnostic to detect male meiotic defects [31–37]. As shown in Fig 2D, wild-type round spermatids contained an equal-sized phase light nucleus and a phase dark nebenkern at a ratio of 1:1 per cell within the cyst of 64 cells. In contrast, *cpes* mutant round spermatids showed 4 regularly sized nuclei (tetranucleate) and 1 abnormally large nebenkern, indicating a failure in cytokinesis in both meiosis I and II divisions (Fig 2E). Quantification of

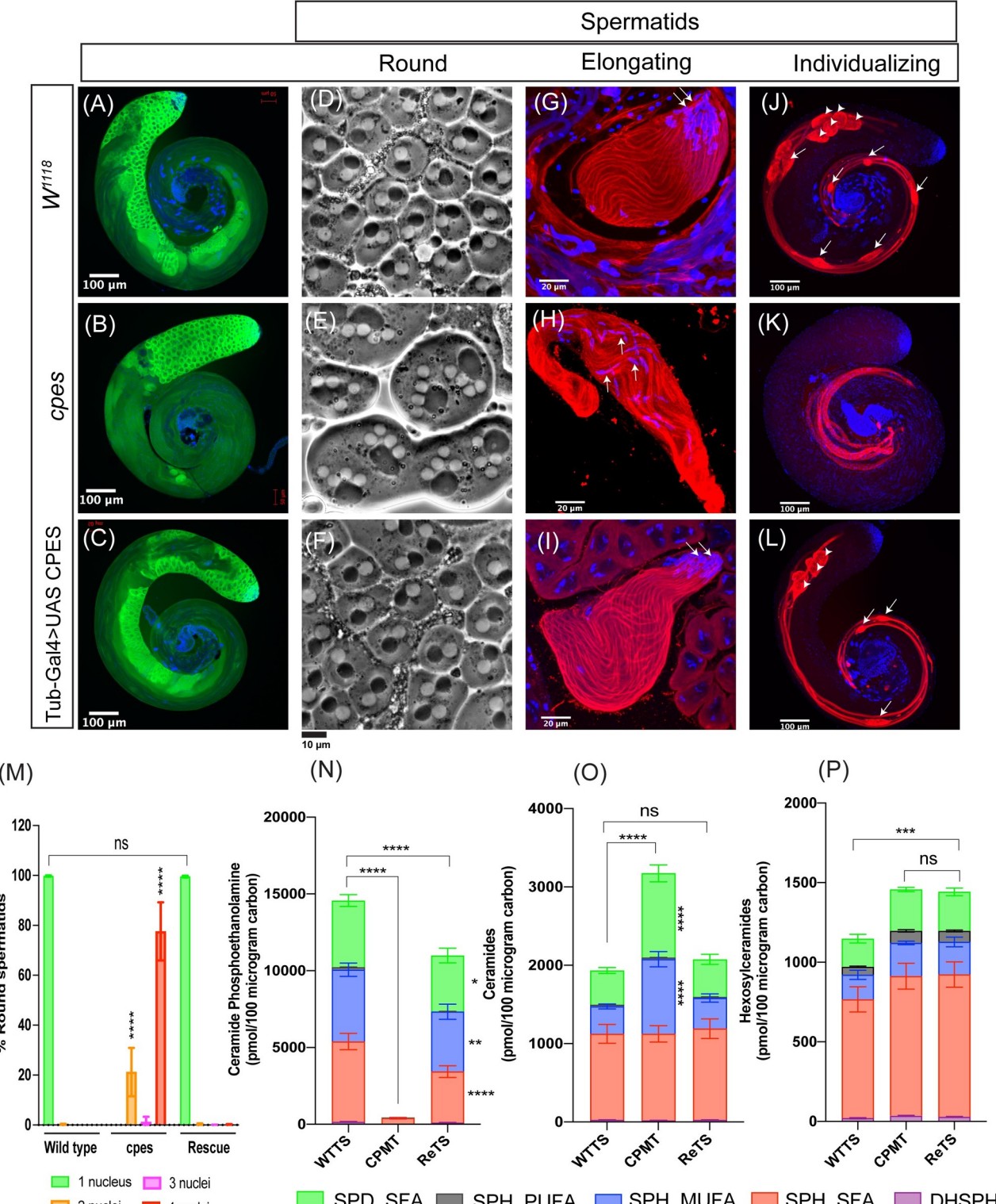

**Fig 2. Cytological, immunofluorescence, and sphingolipid analyses in cpes mutants.** (A–C) Whole mount of *Drosophila* testis immunostained for germ cell-specific Vasa protein primary antibody and Alexa Fluor 488 secondary antibody (green) and DNA with DAPI (blue) to visualize various stages of germ cells including stem cells, spermatogonia, and spermatocytes (A) w[1118], (B) cpes mutant, and (C) Tub-Gal4>UAS-CPES rescue. (D–F) Phase contrast microscopy of live testis squash preparations to visualize round/onion stage spermatids (D) w[1118], (E) cpes mutant, and (F) Tub-Gal4>UAS-CPES rescue. (G–I) Testis squash preparation followed by fixation and immunostaining to visualize early elongating spermatids. Arrow

indicates spermatid head stained with DAPI (blue) for DNA. The beta tubulin primary antibody and Alexa Fluor 568 conjugated secondary antibody (red) were used for immunostaining spermatid tails (axonemes), w[1118] (G), cpes mutant (H), and Tub-Gal4>UAS-CPES rescue (I). (J–L) Whole mount *Drosophila* testis immunostained with cleaved caspase-specific DCP-1 primary antibody and Alexa Fluor 568 secondary antibody (red) to visualize elongated and individualizing spermatids. DAPI stains the DNA (blue). Cystic bulges are shown with arrows and waste bags with arrow heads (J) w[1118] (K) cpes mutant, and (L) Tub-Gal4>UAS-CPES rescue. (M) Quantification of meiotic cytokinesis defects in cpes mutants. Round spermatid count from 3 independent experiments for w[1118] ($n = 1,600$), cpes ($n = 300$), and from 4 independent experiments for tubulin-Gal4 >UAS-CPES rescue ($n = 2,000$). The data underlying this graph can be found in S2 Data. (N–P) Sphingolipid analysis in lipid extracts of dissected testis, WTTS, CPMT, and bam-Gal4>UAS-CPES Rescue testis (ReTS). (N) Comparison of CPE and its subspecies between w[1118], cpes mutant, and ReTS. (O) Comparison of ceramide and its subspecies between w[1118], cpes mutant, and ReTS. (P) Comparison of hexosylceramide subspecies between w[1118], cpes mutant, and ReTS. Statistical significance was calculated using mean, SD, and N in Prism 8. The 2-way ANOVA multiple comparison was used to calculate *p*-values where **** $p \leq 0.0001$, *** $p \leq 0.001$, ** $p \leq 0.01$, * $p \leq 0.05$, and ns $p > 0.05$. Three independent biological replicates were taken for each sample and 250 pairs of testes were used for lipid extraction from each biological replicate. The data underlying the graphs shown in this figure can be found in S1 Data. CPE, ceramide phosphoethanolamine; CPMT, cpes mutant testis; SD, standard deviation; WTTS, wild-type testis.

this defect suggested 80% of the round spermatids were tetranucleate and about 20% showed 2 nuclei (Fig 2M). We did not observe single nucleated round spermatids in *cpes* mutant testes (Fig 2M). This defect is unique to male meiotic cytokinesis since larval neuroblasts undergoing asymmetrical mitotic division did not show defects in cytokinesis (S1 Movie). Females were largely fertile indicating normal oogenesis. Also, during spermatogenesis, we did not observe any defect in the formation of 16 cell staged spermatogonia/spermatocytes indicating that mitotic cytokinesis, even in the testis of *cpes* mutants, was not compromised.

Following the round/onion stage, the 64 interconnected haploid cells in the cyst become highly polarized with the nuclei localizing to one end and the sperm tails (axonemes) growing in the opposite side. These elongating spermatid nuclei face toward the seminal vesicles and the tails grow toward the germline stem cell hub/tip of the testis. As shown in Fig 2G, wild-type early elongating spermatids had all the nuclei facing one side and their axonemes growing to the opposite side indicating that they are highly polarized. In contrast, all the elongating spermatids in *cpes* mutant cysts appeared sickle cell shaped with the nuclei present in the middle-bulged area (occasionally throughout the cyst) and their tails growing on both sides of the cyst (Fig 2H). This data indicates that spermatid polarity is compromised in *cpes* mutants.

Individualization is the final stage of spermatid differentiation wherein structures called the individualization complex (IC) containing actin cones assemble at the head/rostral end of elongated spermatids and start moving away from the nuclei toward the tail/caudal end of the cyst. As the IC travels, it removes the cytoplasmic contents of the cyst and individualizes each spermatozoon with its own plasma membrane. The extruded cytoplasmic content is collected into a sac-like structure called the "cystic bulge" that forms around the IC. As this process proceeds and the IC and cystic bulge reach the end of the flagella, the actin cones and cytoplasmic contents are extruded in a waste bag, the contents of which are degraded later [38]. Activated caspases have a non-apoptotic role in the cystic bulge and the waste bag of individualizing spermatids and are essential for successful spermatid individualization [39]. Immunostaining of wild-type testis with cleaved *Drosophila* Caspase 1 (DCP-1) antibody showed elongated spermatids undergoing individualization that contained cystic bulges and waste bags (Fig 2J). In contrast, *cpes* mutant testis showed defectively elongated spermatids and was devoid of cystic bulges and waste bags indicating failure in spermatid individualization (Fig 2K).

The meiotic cytokinesis, spermatid polarity, and individualization defects of *cpes* mutants are fully penetrant (100%) and ubiquitous expression of wild-type cDNA copy of CPES completely rescued all male sterility phenotypes (Fig 2C, 2F, 2I, 2L and 2M). As shown in Fig 2F, phase contrast imaging of live testis squash preparations showed 1:1 nebenkern to nucleus ratio indicating rescue of meiotic cytokinesis defect. Further, immunostaining of testis squash preparations with tubulin antibody and DAPI showed that spermatid polarity was completely

restored wherein all the nuclei faced one side and their tails faced the other side of the cyst (Fig 2I). DCP-1 staining showed individualizing spermatids with cystic bulges and waste bags indicating normal individualization (Fig 2L).

To investigate the sphingolipid content in *cpes* mutant, we dissected wild type, *cpes* mutant, and germ cell-specific rescue (*bam*-Gal4> UAS-CPES) testes, sphingolipids were extracted and analyzed by SFC/MS/MS as described in the methods (S1 Data). The mutant and rescue samples were processed at the same time as the wild type for mass spectrometry; hence, the wild-type testis sample is the same as in Fig 1 (S1 Data). There was complete loss of CPE in *cpes* mutant testis (Fig 2N) and expression of CPES only in the germ cells was sufficient to restore CPE to near wild-type levels. Loss of CPE synthesis in *cpes* mutants caused concurrent accumulation of ceramides particularly SPH_MUFA and SPD_SFA species (Fig 2O) and expression of CPES in germ cells reduced ceramide levels. HexCer levels were also slightly increased in *cpes* mutant testis (Fig 2P). Interestingly, expression of CPES in germ cells did not reduce the Hex-Cer levels indicating their levels did not correlate with the *cpes* mutant phenotypes in the testis.

## CPES enzymatic activity is required for spermatogenesis and accumulation of ceramide is not the cause for spermatogenesis defects

To determine whether enzymatic activity of CPES is required for spermatogenesis, we generated an active site mutant of CPES. CPES has a conserved amino acid motif $D(X)_2DG(X)_2$ (A/Y)$R(X)_{8-16}G(X)_3D(X)_3D$ in the CDP-alcohol phosphotransferase (CAPT) domain [40]. The final aspartates of this motif were shown to be essential for catalysis of human choline/ethanolamine phosphotransferase CEPT1 [41,42]. We substituted the last 2 active site aspartates of CPES with alanine residues and subcloned into pUAST vector for generating transgenic flies. The UAS_CPES ($DX_3D$ to $AX_3A$) mutant transgene was expressed in *cpes* mutant background using *bam*-Gal4 to test if spermatogenesis phenotypes could be rescued. As shown in S2A and S2B Fig, the active site mutant did not rescue spermatogenesis phenotypes suggesting a crucial role for CPES enzymatic activity in spermatogenesis.

We next investigated whether accumulation of ceramide at the Golgi is responsible for the observed phenotypes. Ceramide is generated in the membranes of ER via the de novo biosynthetic pathway catalyzed by the rate-limiting enzyme serine palmitoyltransferase (SPT) (Fig 1A). Subsequently, ceramide is actively transported from ER to Golgi via ceramide transfer protein (DCERT) and thus absence of DCERT could prevent ceramide accumulation at the Golgi (Fig 1A) [43]. To investigate whether blocking ceramide transport from ER to Golgi could restore spermatogenesis, we generated *cpes* and *dcert[1]* double mutants and their testes were analyzed for meiotic cytokinesis and spermatid polarity phenotypes. However, as shown in S2C and S2D Fig, *cpes*; *dcert[1]* double mutants did not rescue meiotic cytokinesis and spermatid polarity indicating ceramide accumulation at the Golgi (due to lack of CPES activity in the Golgi) may not be responsible for *cpes* mutant phenotypes. In *Drosophila*, transgenic expression of neutral ceramidase was shown to reduce ceramide levels in vivo [44]. We overexpressed ceramidase (UAS CDase) in germ cells using *bam*-Gal4 in *cpes* mutant background and testes from the resulting progeny were analyzed for the rescue of mutant phenotypes. However, expression of CDase did not rescue meiotic cytokinesis and spermatid polarity (S2E and S2F Fig), suggesting that absence of CPE but not accumulation of ceramide is responsible for the phenotypes.

## Transcriptomic and genetic dissection of spatiotemporal role of CPES in spermatogenesis

Spermatogenesis is a conserved process across various taxa that is facilitated by highly dynamic transcriptome. An earlier study leveraged single cell RNAseq and unsupervised clustering to

identify all the major cell classes of the sperm lineage and validated them with previously studied marker genes [45]. To characterize cell type-specific transcriptional signatures in *cpes* mutants in vivo, we performed total RNA-seq with dissected testis from $w^{1118}$, *cpes* mutant, and *bam*-Gal4>UAS CPES rescue. Using Gene Set Enrichment Analysis (GSEA), we compared our RNAseq results with the top 50 expressed genes from each cell type/cellular status that were previously classified to confirm the presence of germline stem cells (GSCs), spermatogonia, spermatocytes, and spermatids (germ cells) as well as cyst stem cells, terminal epithelial cells, and hub cells (somatic cells) [45]. We found that gene sets corresponding to GSC, early spermatogonia, and late spermatogonial cells were enriched in *cpes* mutants compared to wild type and rescue testis (S3 Fig). In contrast, gene sets corresponding to early and late spermatocytes, early, and mature spermatids were enriched in wild type and rescue testis compared to *cpes* mutant testis (S3 Fig). These results suggested that CPES might play important roles during late spermatogonial, early, and late spermatocyte stages for successful completion of spermatogenesis.

To determine the stage of germ cell differentiation at which CPES expression is important, we performed cell type/stage-specific rescue experiments using UAS/Gal4 system in the *cpes* mutant background (Fig 3A). We first expressed UAS-CPES in cyst cells using C587-Gal4 to investigate a possible non-cell autonomous role. C587-Gal4 was expressed in all early somatic cells at the apical tip of the testis including somatic cyst stem cells and cyst cells (S4D and S4H Fig) [46]. As shown in Fig 3, expression of wild-type CPES in cyst cells did not rescue meiotic cytokinesis, spermatid polarity, or individualization defects (Fig 3B, 3F and 3J, respectively). We next expressed wild-type UAS-CPES in germ cells using 3 different Gal4 drivers that have been previously shown to be specifically expressed in stem cells (*nos*-Gal4), transit amplifying spermatogonial cells (*bam*-Gal4), and early spermatocytes (*chif*-Gal4) (Fig 3A). Na*nos* (*nos*) gene encodes for an RNA-binding protein involved in the formation of translational repressor complex. It is required for germ plasm organization, germline development, germline stem cell renewal, and neuronal morphogenesis [47–49]. *nanos*-Gal4-VP16 has been routinely used to specifically express a transgene of interest in male and female germline stem cells. This construct consists of 700 bp *nos* promoter, Gal4-VP16 ORF, *nos* 3′ UTR, and 500 bp of 3′ genomic *nos* transcription unit. In male germ line *nos*-Gal4 primarily expresses in GSC and early spermatogonial stages [50]. Bag of marbles (*bam*) gene encodes for a fusome (a germ cell-specific organelle)-associated protein. The Bam protein is required for activation of the switch from spermatogonia to spermatocytes. The *bam*-Gal4-VP16 construct has been successfully used to drive transgene expression in late spermatogonia and early spermatocytes in *Drosophila*. This construct consists of 900 bp bam promoter, 500 bp *bam* 5′UTR, Gal4:VP16 ORF, and HSP70 3′UTR. Although transgenes driven with *bam*-Gal4-VP16 express in late spermatogonia through early spermatocytes, the peak expression was shown to occur in spermatocytes [51]. Gene trap Gal4 insertion collection screen identified *Chiffon*-Gal4 to drive transgene expression specifically in early spermatocytes and somatic cysts cells [52]. To verify the expression pattern of these Gal4 drivers in our experimental settings, we have crossed *nos*-Gal4 (BDSC#4937) with various pUASP/pUAST-GFP-tagged proteins including pUASP-*alpha-Tubulin*-GFP, pUAST-EGFP, and pUAST-PLCδ-PH-EGFP. As expected, *nos*-Gal4 expression is restricted to early germ cells present at the tip of the testis including GSCs and early spermatogonial cells (S4A, S4E, and S4I Fig). On the contrary, *bam*-Gal4 expression is strong just below the testis tip where spermatogonia and early spermatocytes are expected to be present (S4B, S4F and S4J Fig). However, depending on the stability of the expressed protein, *bam*-Gal4-mediated expression is either restricted to spermatogonia and early spermatocytes (e.g., PLCδ-PH-EGFP, S4J Fig) or persisted through later stages including early, later spermatocytes, and even in spermatids (e.g., *Alpha tubulin* and EGFP, S4B and S4F Fig). Expression of *chif-*

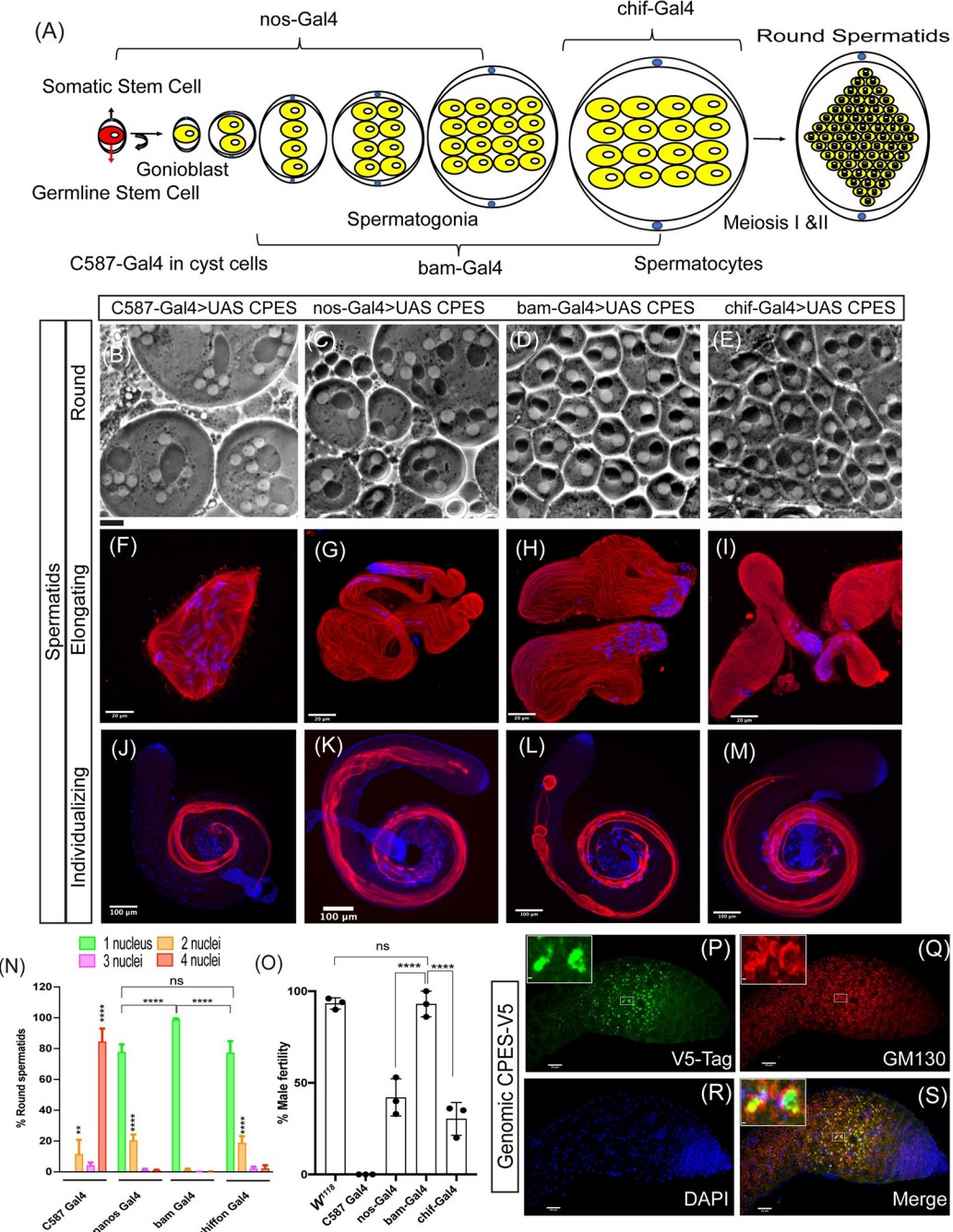

**Fig 3. Genetic dissection of spatiotemporal role of CPES in spermatogenesis.** (A) Schematic illustration of early germ cell differentiation and stage-specific Gal4 expression patterns. (B–E) Phase contrast microscopy of round spermatids in live testis squash preparations. (F–I) Early elongating spermatids immunostained for beta tubulin primary antibody (red) and DAPI for DNA (blue). (J–M) Individualizing spermatids immunostained for cleaved caspase DCP-1 antibody (red) and DAPI for DNA (blue). (B, F, and J) C587-Gal4 drives the expression of UAS-CPES in cyst cells of cpes mutants. (C, G, and K) Nanos-Gal4 drives the expression of UAS-CPES in GSCs and spermatogonia. (D, H, and L) bam-Gal4 drives the expression of UAS-CPES in 4–16 cell stages of spermatogonia and spermatocytes. (E, I, and M) chif-Gal4 drives the expression of UAS-CPES in early spermatocytes. (N) Measurement of cytokinetic defects in testis squash preparations in cyst and germ cell-specific rescue samples. Round spermatid count from 4 independent experiments for C587-Gal4 >UAS-CPES (*n* = 500), 6 independent experiments nos-Gal4 >UAS-CPES (*n* = 2,500), 5 independent experiments for bam-Gal4 >UAS-CPES (*n* = 2,500), and 6 independent experiments for chif-Gal4 >UAS-CPES (*n* = 3,000). (O) Male fertility test where individual male from each of the UAS-CPES rescue experiment was crossed with 3 wild-type females. The vials that produced at least 5 pupae or progeny were counted as fertile and plotted as percent fertility compared to

wild-type males. The data underlying the graphs shown in this figure can be found in S2 Data. (P–S) Whole mount immunostaining followed by confocal imaging of genomic CPES flies where CPES is tagged with V5 at the C-terminus (red). GM130 is a *cis*-Golgi marker (green) and DAPI (blue) stains the DNA. CPE, ceramide phosphoethanolamine; GSC, germline stem cell.

Gal4 was detectable in spermatocytes (S4C Fig) but more strongly in elongated spermatids (S4G and S4K Fig). Interestingly, as shown in Fig 3, we observed varying degree of rescue, depending on the germ cell differentiation stage at which UAS-CPES was expressed. The *nos*-Gal4-dependent expression of UAS-CPES partially rescued both meiotic cytokinesis (Fig 3C and 3N) and spermatid polarity (Fig 3G); however, individualizing spermatids lacked cystic bulges and waste bags indicating individualization was not fully rescued (Fig 3K). The *bam*-Gal4-dependent expression of UAS-CPES in transit-amplifying cells completely rescued meiotic cytokinesis (Fig 3D and 3N) and spermatid polarity and individualizing spermatids contained cystic bulges and waste bags (Fig 3H and 3L). In contrast, only partial rescue of meiotic cytokinesis was observed when UAS-CPES was expressed in spermatocytes using *chif*-Gal4 (Fig 3E and 3N). Although spermatid polarity was completely rescued, the individualizing spermatids largely lacked cystic bulges and waste bags indicating that individualization was defective (Fig 3I and 3M).

To determine the extent to which male fertility was rescued by cell type-specific CPES expression compared to wild type, we performed male fertility tests. As shown in Fig 3O, expression of UAS CPE in transit amplifying spermatogonial cells and spermatocytes (*bam*-Gal4) rescued male fertility to wild-type levels. In contrast, partial rescue was observed when UAS-CPES was expressed only in stem cells (*nos*-Gal4) or in spermatocytes (*chif*-Gal4). Expression of UAS-CPES in cyst cells (C587-Gal4) did not rescue male sterility as all males were sterile (Fig 3O).

To determine in vivo CPES localization more precisely at the protein level, we have generated a construct with a 6,437 bp extended CPES genomic fragment that was modified by recombineering to carry V5-tag at the C-terminus [27]. Transgenic flies expressing genomic CPES tagged with V5 at the C-terminus fully rescued lethality, photosensitive epilepsy, and all other phenotypes including spermatogenesis defects. Immunostaining followed by confocal imaging showed that CPES-V5 was highly expressed in the cells that are present just below the tip of the testis (Fig 3P and 3S). This area of the testis is known to be enriched in late spermatogonia and early spermatocytes [29]. Taken together, these results show that testis-specific CPE generation by CPES from late spermatogonia stage to spermatocyte stage is crucial for successful meiotic cytokinesis, spermatid polarity, and spermatid individualization.

## Insect CPE synthases but not sphingomyelin synthase rescue male sterility defects

Previously, we have shown that CPE has an important structural role in cortex glial plasma membranes [27]. The cortex glial plasma membranes are severely defective in *cpes* mutants leading to loss of cortex glia and neuronal cell body interactions during development and increased susceptibility to light inducible epilepsy in adults. Further, we have shown that overexpression of human sphingomyelin synthase 1 (hSMS1) that produces SM instead of CPE was sufficient to rescue cortex glia plasma membrane defects via establishment of detergent resistant membranes. We wondered whether overexpression of hSMS1 could also rescue the spermatogenesis phenotypes of *cpes* mutant. To this end, UAS-hSMS1 was ubiquitously expressed using tubulin-Gal4 in *cpes*, the adult male testes were dissected, and live testis squash preparations were observed using phase contrast microscopy. However, as shown in S5A Fig,

hSMS1 overexpression did not rescue male meiotic cytokinesis. Further, immunostaining analysis showed that spermatid polarity was also not rescued (S5E Fig), suggesting an important role for the CPE head group in spermatogenesis. We next investigated whether expression of other insect-derived CPES could rescue *cpes* mutant phenotypes. To test this, we ubiquitously expressed *Aedes aegypti* (yellow fever mosquito, amino acid sequence shows 60% identity and 77% similarity) and *Bombyx mori* (domestic silk moth, amino acid sequence shows 43% identity and 58% similarity) derived UAS-CPES homologs in *cpes* mutant background using tubulin-Gal4. As shown in S5 Fig, mosquito and silk moth CPES were able to completely rescue defects in meiotic cytokinesis (S5B–S5D Fig) and spermatid polarity (S5F and S5G Fig, respectively).

To investigate the nature of sphingolipid species synthesized by hSMS1, *A. aegypti*, *B. mori* derived CPES, we specifically expressed these transgenes in germ cells using *vasa*-Gal4, their male reproductive system was dissected, lipids were extracted and subjected to sphingolipid analysis using SFC/MS/MS as described in the methods section (S3 Data). As shown in S5H Fig, germ cell-specific expression of *Drosophila* CPES, *B. mori* CPES completely restored CPE to the wild-type amounts. *A. aegypti* CPES also synthesized significant amount of CPE but to a lower amount than *Drosophila* CPES. Still, all 3 distinct CPE species are synthesized by *A. aegypti* CPES and *Bombyx* CPES (S5H Fig). As anticipated, hSMS1 did not synthesize CPE, but produced significant amount of SM (S5K Fig). However, the amount of SM is relatively low compared to CPE, perhaps due to limited expression/stability of hSMS1 in germ cells (vasa-Gal4). Indeed, ubiquitous expression of hSMS1 using actin-Gal4 driver significantly increases the SM amounts (S5L Fig). However, expression of hSMS1 with ubiquitous drivers like actin-Gal4 and tubulin-Gal4 did not rescue the spermatogenesis defects (S5A and S5E Fig) suggesting the importance of the CPE head group. We next analyzed the ceramide content in these samples and found that ceramide levels were significantly higher in *cpes* mutants and hSMS1 rescue compared to *Drosophila* CPES rescue (S5I Fig). However, as discussed in S2 Fig, higher ceramide content might not contribute to the cytokinetic defects in *cpes* mutants. Similarly, higher amounts of HexCer in *A. aegypti* CPES rescue and *B. mori* CPES rescue also did not correlate with the rescue of spermatogenesis phenotypes (S5J Fig). Taken together, these results suggest that *Drosophila* spermatogenesis is strictly dependent on CPE and the head group of CPE plays an important role.

## Aberrant central spindle behavior but not localization of PIPs at the plasma membrane is responsible for defective meiotic cytokinesis in *cpes* mutants

Precise organization of central spindle microtubules is not only required for initial cleavage furrow formation but also for maintenance of contractile structures during furrow ingression. Several microtubule interacting proteins including Fascetto, kinesin 6 family member MKLP1/Pavarotti (pav), Chromokinesin klp3A, Orbit, etc., are enriched in the central spindle midzone and are essential for cytokinesis [13,53]. To determine the transcriptional signatures relating to spindle organization and function in our RNAseq data, we performed GSEA pathway comparisons between $w^{1118}$ (WT), *cpes* (KO), and rescue (RES) samples. As shown in S6A Fig, we found that several pathways including spindle organization, metabolism, endocytosis, signaling, male meiotic cytokinesis, spermatid differentiation, sperm individualization, etc., were significantly altered in *cpes* mutants (S6A Fig). Red bars in S6A Fig indicate gene sets enriched in *cpes* mutants and blue bars indicate gene sets enriched in WT and RES. GSEA showed many genes involved in spindle elongation and spindle organization were positively correlated with the mutant phenotype (S6B and S6E Fig). The heatmap comparison for genes

involved in spindle organization and elongation are shown (S6B–S6H Fig). Several of the genes implicated in spindle organization have also been annotated under those for spindle elongation and hence appear under both categories. However, as shown in S3 Fig, relative enrichment of earlier stages compared to later stages limits the accurate prediction of altered pathways in *cpes* mutants. Notably, immunostaining analysis of testis squash preparations with alpha tubulin antibody did not show consistent differences in astral spindle organization and chromosomal alignment in metaphase and anaphase spermatocytes (S7A–S7C, S7G and S7H Fig). However, spindle microtubules and central spindle (marked by feo-Cherry) were less dense and disorganized in early to late telophase spermatocytes indicating spindle behavior is affected in *cpes* mutants during cytokinesis (S7D–S7F and S7J–S7L Fig).

To better study spindle behavior during male meiotic cytokinesis in vivo, we performed live imaging on isolated cysts. We first dissected the testis in M3 insect cell culture media and cut open the muscle sheath to release intact spermatocyte cysts into the media. Subsequently, we transferred spermatocyte cysts to poly-D-lysine-coated cover glass dishes and performed live imaging using Andor spinning disk confocal microscopy as described in the methods section. We chose mCherry tagged microtubule cross linking protein Feo (Ubi-p63E-*feo*-mCherry) as a marker for central spindle. Feo-mCherry was shown to accumulate at the anaphase B and telophase central spindle [54]. We also used spaghetti squash (sqh) gene encoding myosin II regulatory light chain (RLC) tagged to GFP (*sqh*-GFP-RLC) as a marker for the contractile ring, an actin-myosin-based structure, that assembles during late anaphase [55,56]. The actin and myosin ring positioned midway between 2 spindle poles is thought to drive the formation of the cleavage furrow during telophase. As shown in S2 Movie and Fig 4A, the actomyosin ring (*sqh*-GFP-RLC) quickly assembles around the central spindle during late anaphase and both simultaneously constrict through early to late telophase. At the end of late telophase *feo*-mCherry disassembles, however, *sqh*-GFP-RLC remains and becomes part of ring canal/cytoplasmic bridges between daughter cells (S2 Movie). Interestingly, in *cpes* mutants, the initial assembly of central spindle and actomyosin ring occurred normally (S3 Movie); however, central spindle and actomyosin ring constriction became uneven since there was barely any furrow ingression and led to substantial disengagement between the contractile ring and the central spindle during meiosis (Fig 4A, 4C and 4D). As a result, in the mutants the actomyosin ring either detached from the plasma membrane and/or the central spindle and actomyosin ring disassembled prematurely (S3 Movie). Quantification of this behavior in live *cpes* mutant spermatocytes suggested that more than 80% of the spermatocytes showed disengagement of the contractile ring and central spindle prior to contractile ring detachment from the plasma membrane (Fig 4B and S3 Movie). Other minor variations included contractile ring detachment from the plasma membrane prior to central spindle and contractile ring disengagement, contractile ring sliding and dumping to one end, and the contractile ring and central spindle being destabilized simultaneously at one end (Fig 4B). Under similar experimental settings in intact cysts, cytokinetic defects are negligible. Expression of UAS-CPES in germ cells was sufficient to rescue this phenotype (S4 Movie and Fig 4A, 4C and 4D).

Previous studies have shown that phosphatidylinositols (PIP) play critical roles in somatic and meiotic cytokinesis [57]. Further, in mammalian (HeLa) cells undergoing mitosis, sphingomyelin-rich lipid domains in the outer leaflet of the cleavage furrow were required for accumulation of $PI(4,5)P_2$ to the cleavage furrow which in turn was required for proper translocation of RhoA and progression of cytokinesis [10]. We wondered whether the distribution of PIP at the plasma membrane or at the cleavage furrow is altered in *cpes* mutants during meiosis. To investigate PIP distribution in *cpes* mutant membranes, we used $PI(4,5)P_2$ reporter UAS-PLCδ-PH-EGFP (UAS regulatory sequence drives expression of EGFP fused to the PH domain of human PLCdelta 1) and $PI(3,4,5)P_3$ reporter tGPH (an alphaTub84B promoter

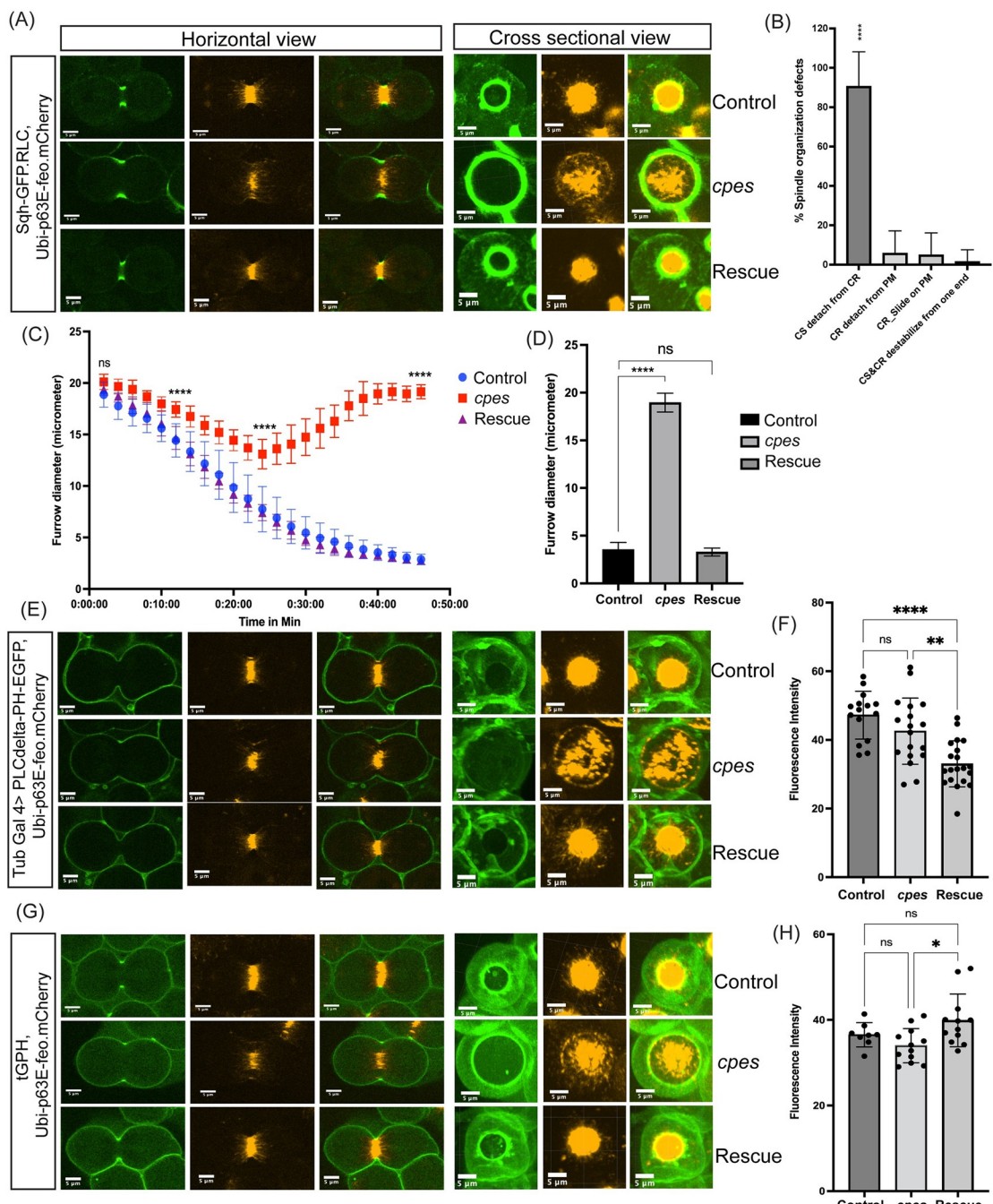

**Fig 4. Aberrant central spindle behavior but not localization of PIPs at the plasma membrane is responsible for meiotic cytokinesis defect in cpes mutants.** (A, C, and D) Snapshots of live spermatocytes undergoing meiosis I cytokinesis. In horizontal view (left), one slice of a z-stack is shown. In cross-sectional view across cleavage furrow, maximum intensity projection image is shown (right). Control spermatocytes (top row), cpes mutant spermatocytes (middle row), and tubulin-Gal4>UAS-CPES rescue (bottom row). (A) Spermatocytes expressing myosin II RLC tagged to GFP under its own promoter (sqh) that localizes to contractile ring and mCherry tagged fascetto (feo) under the control of Ubi-p63E promoter that binds to central spindle microtubules. (B) Quantification of spindle organization defects in live cpes mutant spermatocytes. First observable phenotype was measured and plotted. CS detachment from CR, detachment of CR from PM, CR slides on PM and CS and CR destabilizing from end. (C) Quantification of rate and magnitude of cytokinetic furrow ingression using ImageJ and Prism9. Cytokinetic furrow diameter was measured as a function of time in dividing spermatocytes from control ($n$ = 12), mutant ($n$ = 10), and rescue sample ($n$ = 9). Statistics were performed using 2-way ANOVA multiple comparisons in Prism 9. (D) A representative bar diagram for cytokinetic furrow diameter at 40 min post furrow ingression in C. (E) Spermatocytes expressing GFP-tagged PH domain of PLCD (UAS-PLCδ-PH-EGFP) under the control of Tub-Gal4 that localizes to PM in the presence of phosphatidylinositol-

4,5-bisphosphate (PI(4,5)P2) and Ubi-p63E-feo-mCherry. (F) Fluorescence intensity measurements for live spermatocyte cysts expressing Tub-Gal4> UAS-PLCδ-PH-EGFP, control ($n = 15$ cysts), cpes mutants ($n = 18$ cysts), and rescue ($n = 20$ cysts). Statistics were performed using ordinary 1-way ANOVA multiple comparisons in Prism 9. (G) Spermatocytes expressing GFP-tagged PH domain of steppke/Grp1 under the control of alpha Tub84B promoter (tGPH) that localizes to PM in the presence of phosphatidylinositol-3,4,5-triphosphate (PI(3,4,5)P$_3$) and Ubi-p63E-Feo-mCherry. (H) Fluorescence intensity measurements for live spermatocyte cysts expressing tGPH; control ($n = 8$ cysts), cpes mutants ($n = 12$ cysts), and rescue ($n = 12$ cysts). Statistics were performed using ordinary 1-way ANOVA multiple comparisons in Prism 9. The data underlying the graphs shown in this figure can be found in S2 Data. CS, central spindle; CR, contractile ring; PH, pleckstrin homology; PIP, phosphatidylinositol phosphate; PM, plasma membrane; RLC, regulatory light chain.

drives the expression of EGFP fused to the PH domain of Steppke) [58–60]. The PI(4,5)P$_2$ reporter PLCδ-PH-EGFP was shown to uniformly localize to the entire membrane including cleavage furrow in *Drosophila* spermatocytes [61]. Indeed, we have seen similar uniform distribution in both control and *cpes* mutant spermatocyte plasma membranes indicating normal distribution of PI(4,5)P$_2$ (S5 and S6 Movies and Fig 4E). Live imaging analysis on *cpes* mutant spermatocytes expressing PLCδ-PH-EGFP revealed normal cleavage furrow initiation, but progression of furrow failed due to the disconnect between central spindle and the furrow (S6 Movie and Fig 4E). Expression of wild-type CPES rescued this phenotype (S7 Movie and Fig 4E). Of note, PLCδ-PH-EGFP localization to spermatocyte plasma membrane was significantly reduced upon loss of cyst cells although this acute reduction did not prevent spermatocytes from completing cytokinesis during meiosis 1. Fluorescence intensity measurements for PLCδ-PH-EGFP localization showed no significant difference between control, and *cpes* mutant cysts suggesting PI(4,5)P$_2$ localization was not affected in *cpes* mutants (Fig 4F). Although, rescue cysts showed reduction in PI(4,5)P$_2$ levels, it did not correlate with the rescue phenotype. We next investigated the PI(3,4,5)P$_3$ distribution in control meiotic spermatocytes using tGPH. Interestingly, we found that tGPH localized to plasma membranes and became significantly enriched at cleavage furrows (S8 Movie and Fig 4G). Of note, even here, we have observed that plasma membrane localization and cleavage furrow enrichment of tGPH strictly depended on the intact cyst. While it is currently unknown if PI(3,4,5)P$_3$ localization to the spermatocyte cleavage furrows has any impact on cytokinesis, a recent study has shown that cytohesin Steppke reduces tissue tension by inhibiting the actomyosin activity at adherens junctions and actomyosin network assembly is necessary and sufficient for local Steppke accumulation in *Drosophila* embryos [62]. Next, we wondered if cleavage furrow enrichment of PI(3,4,5)P$_3$ is altered in *cpes* mutants. However, as shown in S9 Movie and Fig 4G, PI(3,4,5)P$_3$ localization to the cleavage furrow was not altered but again there was a disconnect between cleavage furrow and central spindle during cleavage furrow ingression. Expression of wild-type CPES rescued this phenotype (S10 Movie and Fig 4G). Further, fluorescence intensity measurements showed no significant difference between control and *cpes* mutant cysts, suggesting PI(3,4,5)P$_3$ distribution is not compromised in *cpes* mutants (Fig 4H). Overall, these observations suggest that aberrant spindle behavior, but not the localization of PIPs to the plasma membrane or cleavage furrow, is responsible for meiotic cytokinesis defect in *cpes* mutants.

## CPE endocytosis and targeting to cleavage furrows occurs during meiotic cytokinesis

To determine how CPE promotes the synergy between central spindle constriction and cleavage furrow ingression during cytokinesis, we have utilized a recombinant CPE-binding protein tagged with mCherry as a reporter for visualizing CPE on spermatocyte plasma membranes. The mushroom-derived protein of the aegerolysin family pleurotolysin A2 (PlyA2) was shown

to specifically bind to CPE and was demonstrated to be a versatile tool for visualizing CPE in live *Drosophila* tissues [63]. We have reevaluated the efficacy of recombinant PlyA2-mCherry protein in detecting endogenous CPE on live spermatocyte membranes. The wild type and *cpes* mutant spermatocytes were treated with 10 μg/ml of PlyA2-mCherry in M3 insect media for 1 h at room temperature, washed with fresh M3 media, and imaged on an Andor spinning disk confocal microscope. As shown in Fig 5A–5D, PlyA2-mCherry specifically bound to the plasma membranes of wild-type spermatocytes but not to the *cpes* mutant spermatocytes showing specificity of this protein binding to CPE. Interestingly, we found that PlyA2-m-Cherry was endocytosed in wild-type spermatocytes (Fig 5A). To determine the nature of the endosomes where CPE is enriched, we performed live imaging on spermatocytes expressing endogenous EYFP-MYC-tagged Rab proteins [64]. The small GTPases Rab4, Rab7, and Rab11 were shown to mark early, late, and recycling endosomes, respectively [65–67], while Rab6 was shown to be associated with the Golgi membranes [68]. The spermatocytes expressing individual EYFP-Rab proteins were treated with recombinant PlyA2-mCherry and imaged. As shown in Fig 5G, 5M and 5P, PlyA2-mCherry colocalized with Rab4, Rab7, and some of the Rab11 endosomes, respectively. Colocalization coefficient measurements further support these observations (Fig 5Q). However, PlyA2-mCherry containing structures did not colocalize with Rab 6 positive Golgi membranes (Fig 5J and 5Q). These results suggested that CPE on spermatocyte plasma membranes is actively targeted to the endocytic pathway. We next wondered whether endosomes carrying CPE migrate to cleavage furrows during meiotic cytokinesis. EYFP-Rab 7 and EYFP-Rab11 were strongly expressed in spermatocytes undergoing meiosis compared to Rab4 and therefore we followed their dynamics during meiotic cytokinesis. Strikingly, as shown in Fig 6A–6G and S11 Movie, PlyA2-mCherry containing Rab7 positive endosomes were actively targeted to cleavage furrows. Similar EYFP-Rab7 localization was seen in intact cysts (Fig 6H–6J). PlyA2-mCherry also colocalized to some of the EYFP-Rab11 recycling endosomes which in turn localized to ingressing membranes at the cleavage furrow (Fig 6K–6Q and S12 Movie). Overall, the endocytosis of CPE and targeting to the ingressing cleavage furrow via Rab7 and Rab11 marked endosomes indicates their importance in delivery of lipids to the ingressing membranes of the furrow.

To further verify the functional significance of endosome mediated CPE trafficking in cytokinesis, we performed genetic experiments involving conditional expression of EYFP-Rab11DN (BDSC#23261) protein, *rab35* and *rab7* mutants [69]. Rab11 null mutants die early during development [22]. In that study, it was demonstrated that when spermatocytes enter meiosis 1 and as the Golgi disassembles, Rab11 associated with the ER compartment. Rab11 also localized to vesicles at the cell poles during anaphase and early telophase and at the cell equator during mid- and late telophase. Larval gonads of escaper flies carrying Rab11 semilethal transheterozygote alleles showed abnormal constriction of the contractile rings and a failure of cytokinesis in 10% to 30% of the flies. We chose to conditionally express dominant negative EYFP-Rab11$^{S25N}$ ubiquitously using Gal80$^{ts}$ and tub-Gal4 to study cytokinetic defects in spermatocytes. The expression of EYFP-Rab11S25N (Rab11 DN) in late third instar larvae was induced by incubation at 30°C for 4 days. The late pupal testes were dissected, and cytological analysis was performed to measure cytokinetic defects. Consistent with previously published results [22,24], we found significant increase in cytokinetic defects in Rab11DN expressing gonads (S8B and S8F Fig), confirming important roles for Rab11 mediated membrane trafficking during cytokinesis. To further investigate CPE positive endosome behavior during cytokinesis, we treated Rab11 DN expressing spermatocytes with PlyA2-mCherry and performed live cell imaging. Interestingly, compared to control spermatocytes (S9A–S9C and S9M Fig and S13 Movie), Rab11 DN expressing spermatocytes showed significant reduction in CPE positive endosome localization to cleavage furrow (S9D–S9F and S9M Fig and S14

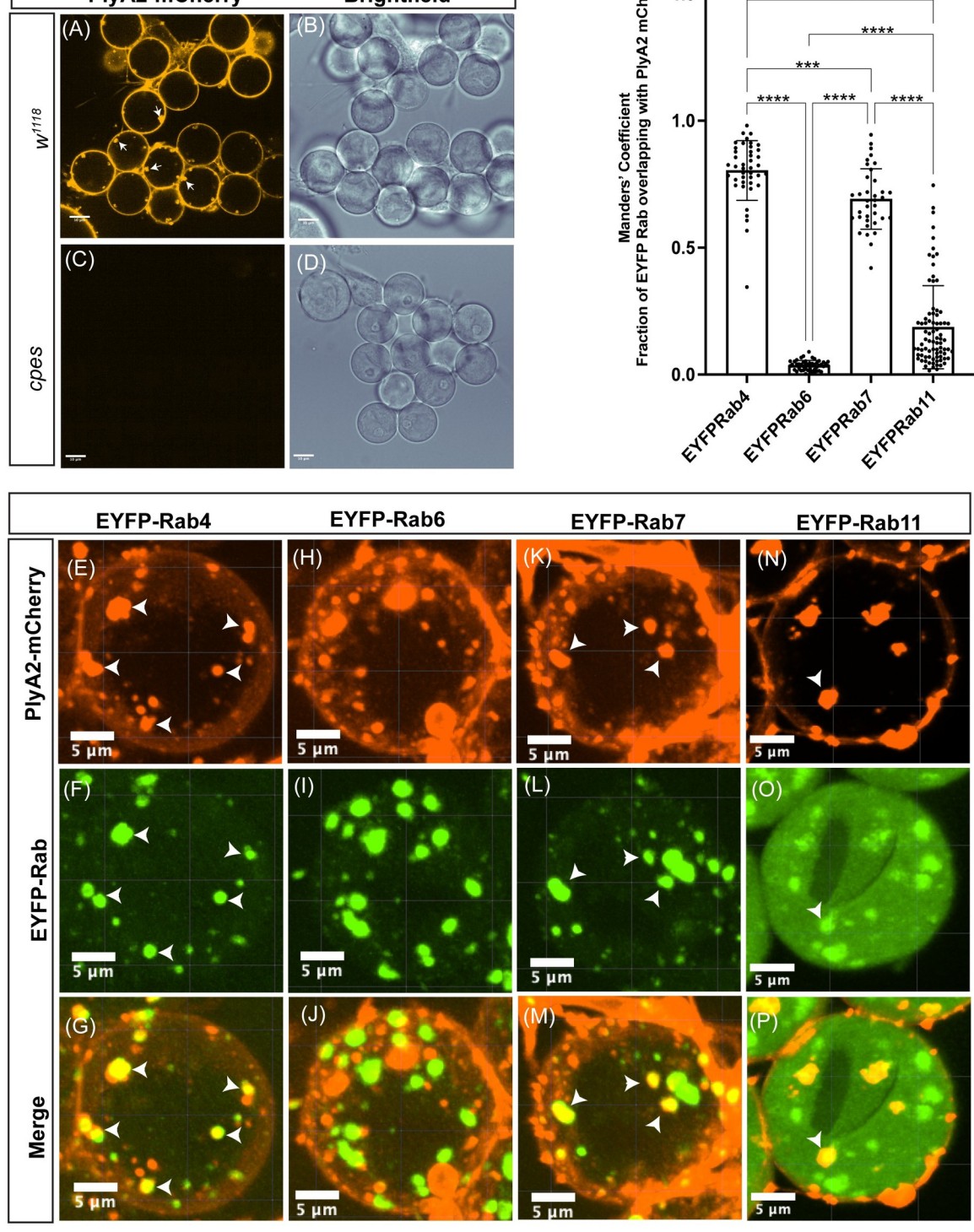

**Fig 5. CPE localizes to plasma membrane and endosomes.** (A–D) Live spermatocytes were incubated with purified PlyA2-mCherry (10 μg/ml) for 30 min followed by washes with M3 insect media (2×) and imaged using spinning disk confocal microscopy. (A and B) $w^{1118}$ spermatocytes, (C and D) cpes spermatocytes. (E–P) Live spermatocytes expressing endogenous EYFP tagged Rab proteins were incubated with PlyA2-mCherry for 1 h followed by washes with M3 insect media (2×) and imaged using spinning disk confocal microscopy. (E–G) EYFP-Rab4 spermatocytes, (H–J) EYFP-Rab6 spermatocytes, (K–M) EYFP-Rab7 spermatocytes, (N–P) EYFP-Rab11 spermatocytes. Arrowheads indicate colocalizing structures. (Q) Colocalization coefficient (Mander's coefficient) measurements showing fraction of EYFP Rab proteins colocalize with internalized PlyA2-mCherry. Mander's coefficient was determined using ImageJ

plugin JACoP. Each dot in the graph represents a 2D image consisting of 8–16 spermatocytes, Rab4 ($n = 42$), Rab6 ($n = 47$), Rab7 ($n = 36$), and Rab11 ($n = 81$). Statistical significance was calculated using mean, SD, and N in Prism 9. The ordinary 1-way ANOVA multiple comparison was used to calculate $P$ values where $****P \leq 0.0001$, $***P \leq 0.001$, $**P \leq 0.01$, $*P \leq 0.05$, and ns $P > 0.05$. The data underlying the graph shown in this figure can be found in S2 Data. CPE, ceramide phosphoethanolamine; SD, standard deviation.

Movie). Further, cleavage furrow regressed following ingression in Rab11 DN expressing spermatocytes (S14 Movie). These results suggest that normal Rab11 function is required for furrow-specific localization of CPE containing endosomes. Cleavage furrow regression after complete ingression in Rab11 DN expressing spermatocytes indicates endosome mediated CPE addition may play an important role in late steps of cytokinesis perhaps by providing stability to the ingressed membranes.

Rab35 is another recycling endosomal marker that was shown to be essential for cytokinesis in mitotic cells [70]. A recent study showed that Rab35 mutant flies are semilethal and mutant male escapers were observed [71]. We performed cytological and immunostaining analysis on Rab35 mutant testes and found significant cytokinetic defects, spermatid polarity, and individualization defects (S8C and S10A–S10D Figs). Earlier studies have shown that Rab35 localizes to the plasma membrane and endosomal compartments in mitotic cells [70]. However, EYFP-Rab35 did not show clear plasma membrane or endosomal localization in immunostained spermatocytes, instead it was predominantly localized to cytosol (S10E Fig). However, we did see its enrichment on endosomal compartment in cyst cells (S10E and S10F Fig). Interestingly, in the live spermatocytes EYFP-Rab35 shows mitochondria-like localization pattern in spermatocytes undergoing cytokinesis and nebenkern in round spermatids (S10G and S10H Fig, respectively). Nevertheless, we have investigated CPE positive endosome behavior in Rab35 mutant spermatocytes and found no significant difference in endocytosis and localization of CPE containing endosomes to the cleavage furrow (S9G–S9I and S9M Fig and S15 Movie). These results suggest that Rab35 may not be involved in cleavage furrow-specific localization of CPE positive endosomes. However, we cannot rule out the possibility that Rab35 could act after CPE positive endosome translocation to the cleavage furrow, i.e., lipid recycling at the furrow. Another possibility could be that Rab35 is required for delivery of other lipids and or protein cargo such as phosphatidylinositol 5 phosphatase (OCRL), oxidoreductase (MICAL1), etc., [23], independent of CPE, to facilitate late steps of cytokinesis in spermatocytes.

We next investigated the role of Rab7 in CPE positive endosome localization to the cleavage furrow and cytokinesis. Rab7-null mutant flies generated by imprecise excision of P-element resulted in a 1,025 bp deletion that removed most of the protein coding exon and the 5′ untranslated region (UTR) [69]. The mutants displayed late pupal lethality and western blotting with Rab7 antibody confirmed the absence of Rab7 protein in the mutant pupae (S8D Fig). To investigate meiotic cytokinetic defects in Rab7 mutants, we performed testis squash preparations followed by phase contrast microscopy. Interestingly, we found significant increase (15%) in meiotic cytokinesis defects in Rab7 mutants (S8E and S8F Fig) compared to wild-type controls where cytokinetic defects are virtually zero [22,24] (S8A Fig), suggesting a significant role for Rab7 in male meiotic cytokinesis. Previous studies have shown that Rab7 protein is required downstream of late endosomes or multivesicular bodies (MVBs) for transfer of cargo to lysosomes [72]; therefore, formation and migration of late endosomes to the cleavage furrow might not be affected in Rab7 mutants. Consistent with this possibility, we observed that PlyA2-mCherry continued to be endocytosed and targeted to cleavage furrow in Rab7 mutants (S9J–S9L and S9M Fig and S16 Movie). However, presence of significant cytokinetic defects in Rab7 mutants indicate that normal Rab7 function is required for proper dynamics of CPE positive endosomes via as yet unknown mechanisms during cytokinesis.

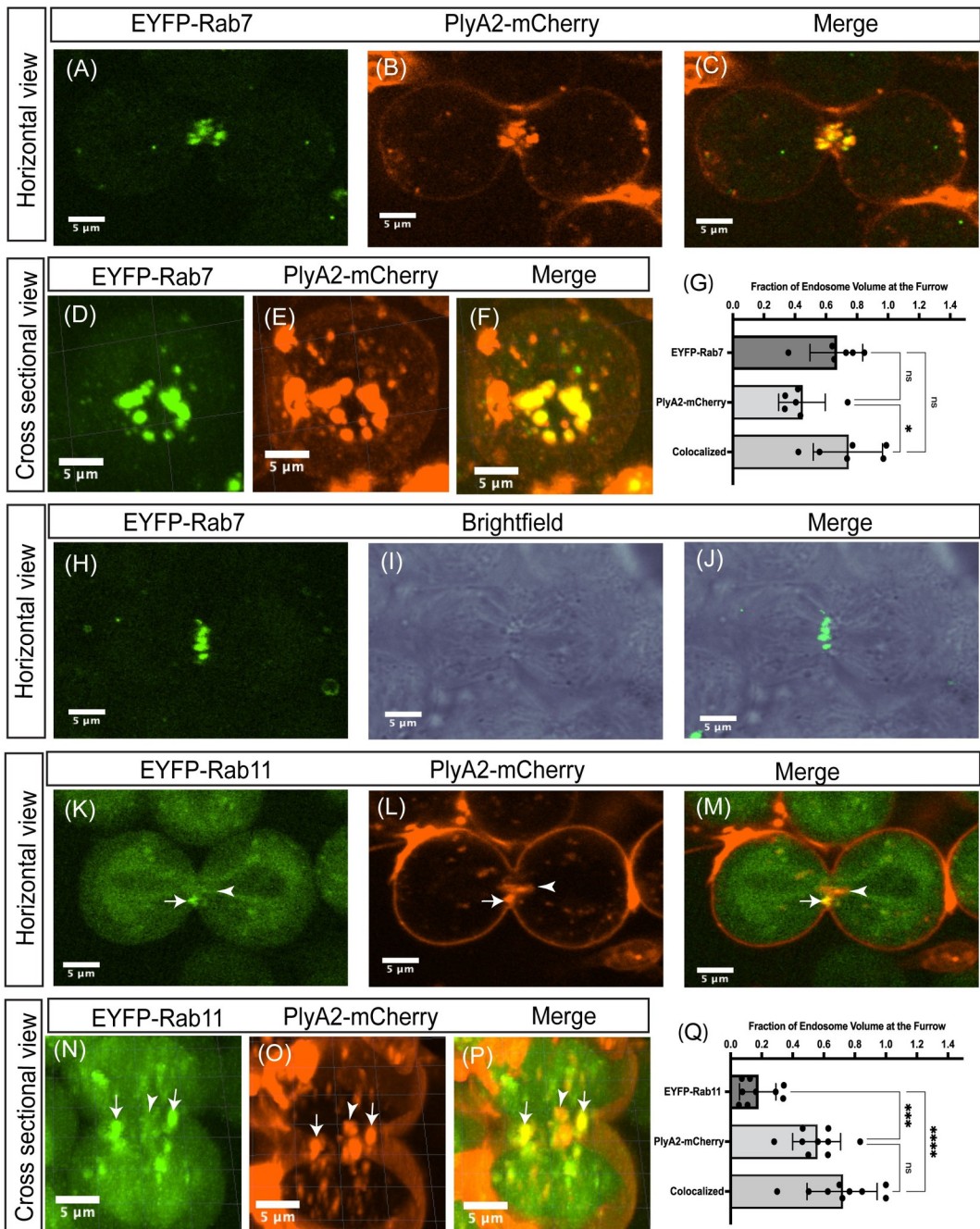

**Fig 6. CPE containing endosomes translocate to cleavage furrows during male meiotic cytokinesis.** (A–F) The EYFP-Rab7 expressing spermatocytes are treated with PlyA2-mCherry and snapshots of spermatocytes undergoing meiosis I cytokinesis are shown. (A–C) Horizontal view of 1 z slice is shown. (D–F) Cross-sectional view of a spermatocyte undergoing cytokinesis, maximum intensity projection image was shown. (G) Quantification of endosomal volume localized to cleavage furrow in EYFP-Rab7 expressing and PlyA2-mCherry treated spermatocytes. Using surfaces tool in Imaris software, we have calculated the total endosomal volume and fraction of which localized to cleavage furrow in 3D images (*n* = 6 spermatocytes from 3 independent experiments). (H–J) Localization of EYFP-Rab7 in intact cysts where spermatocytes are encapsulated by cyst cells. Horizontal view of a single z slice is shown. (K–P) EYFP-Rab11 expressing spermatocytes were treated with PlyA2-mCherry and meiosis I cytokinesis was imaged. (K–M) Horizontal view of single z slice is shown. (N–P) Cross-sectional view, maximum intensity projection image is shown. White arrows indicate colocalization and white arrowheads indicate lack of colocalization. (Q) Quantification of a fraction of endosomal volume localized to cleavage furrow in EYP-Rab11 expressing, PlyA2-mCherry treated spermatocytes. Endosomal volume in 3D images were calculated using surfaces tool in Imaris software (*n* = 9 spermatocytes from 3 independent experiments). The data underlying the graphs shown in this figure can be found in S2 Data. CPE, ceramide phosphoethanolamine.

Taken together, these genetic analyses indicate that Rab7, Rab11, and Rab35 functions are required for normal male meiosis cytokinesis; however, Rab7 and Rab11 functions are more directly linked to CPE positive endosome behavior during cytokinesis.

## Multivesicular bodies deliver CPE laden vesicles to the cleavage furrow

We performed correlative light and focused ion beam scanning electron microscopy (CLEM/FIB-SEM) to gain structural insight into the endosomes that are enriched at the cleavage furrow/midbody during cytokinesis. Spermatocytes expressing EYFP-Rab7 were labeled for endocytosed CPE with PlyA2-mCherry and allowed to progress to meiosis 1. Cysts showing PlyA2-mCherry and EYFP-Rab7 colocalization at the cytokinetic furrow were fixed, imaged by light microscopy, then stained, resin embedded, and prepared for FIB-SEM. Fig 7A–7C shows CLEM-FIB-SEM of a cyst undergoing cytokinesis showing colocalization of Rab7 with endocytosed CPE at the furrow. 3D correlation of the LM and FIB-SEM image volumes allowed us to localize the Rab7/PlyA2 double positive signals to endosomes that appeared as spherical-shaped structures ranging from 300 to 800 nm in diameter. The lumen of these endosomes was filled with intraluminal material which at FIB-SEM resolutions could be discerned, but not definitely resolved, as vesicles packed into the lumenal volume. Although we were able to capture endosomes in the cytokinetic furrow, it was technically difficult to visualize and arrest them in physical association with the ingressing furrow membranes using these flies (S17 Movie). Therefore, we decided to examine cytokinesis in cysts where the contractile ring is in proximity to the ingressing membrane, and the endosomes could be visualized during live imaging.

We examined spermatocytes expressing tubulin-Gal4>mRFP-Anillin that labels the contractile ring and endogenously expressing EYFP-Rab7 that marks late endosomes to capture membrane proximal endosomes (S18 Movie). Anillin (also known as scraps) encodes for a conserved pleckstrin homology domain containing protein that was shown to bind actin, microtubules, and nonmuscle myosin II and required for stabilization of contractile ring [53]. Dividing cysts were imaged, fixed, and cysts with Rab7-Anilin colocalization at the cytokinetic furrow were prepared for FIB-SEM imaging (Fig 7D and 7E). Using CLEM/FIB-SEM, we were able to capture multiple Rab7 positive endosomes docked at the ingressing furrow (Fig 7E–7M). S19 Movie shows the 3D reconstruction of the vesicles captured at the furrow seen in Fig 7K. We observed multiple instances of this docking of endosomes in dividing spermatocytes (Fig 7N). S20 Movie is a sub-volume reconstruction of the dividing cells seen in Fig 7N. While we observed docking of the endosomes to the ingressing membrane in the samples we studied, we were unable to visualize the fusion of vesicles to the ingressing membrane or to the proteinaceous structure that was anchoring the endosomes (Fig 7 and S20 Movie). To visualize the endosomes at the furrow at higher resolution, we carried out CLEM-SEM-TEM of the dividing meiotic spermatocytes and examined vesicles that appeared to be in close proximity to the cleavage furrow (Fig 8). *Drosophila* spermatocyte cysts expressing tubulin-Gal4>mRFP-Anillin and EYFP-Rab7 undergoing meiotic cytokinesis were identified, fixed, and resin embedded (Fig 8A). Serial sectioning of the correlated cysts followed by TEM imaging revealed the structures as MVB-like organelles, filled with intraluminal vesicles varying in size between 20 to 50 nm in diameter (Fig 8B–8D), consistent with previously described multivesicular endosomes (MVEs) or MVBs [73]. Again, rather than seeing evidence of fusion, we found that the outer membrane of several of these endosomes was discontinuous (Fig 8C). Interestingly, the MVBs that showed significant loss of outer membrane integrity appeared to have released intraluminal vesicles in the vicinity of the ingressing membranes (Fig 8D, top left), possibly providing a proximal source of lipid laden vesicles for membrane biogenesis (Fig 8E).

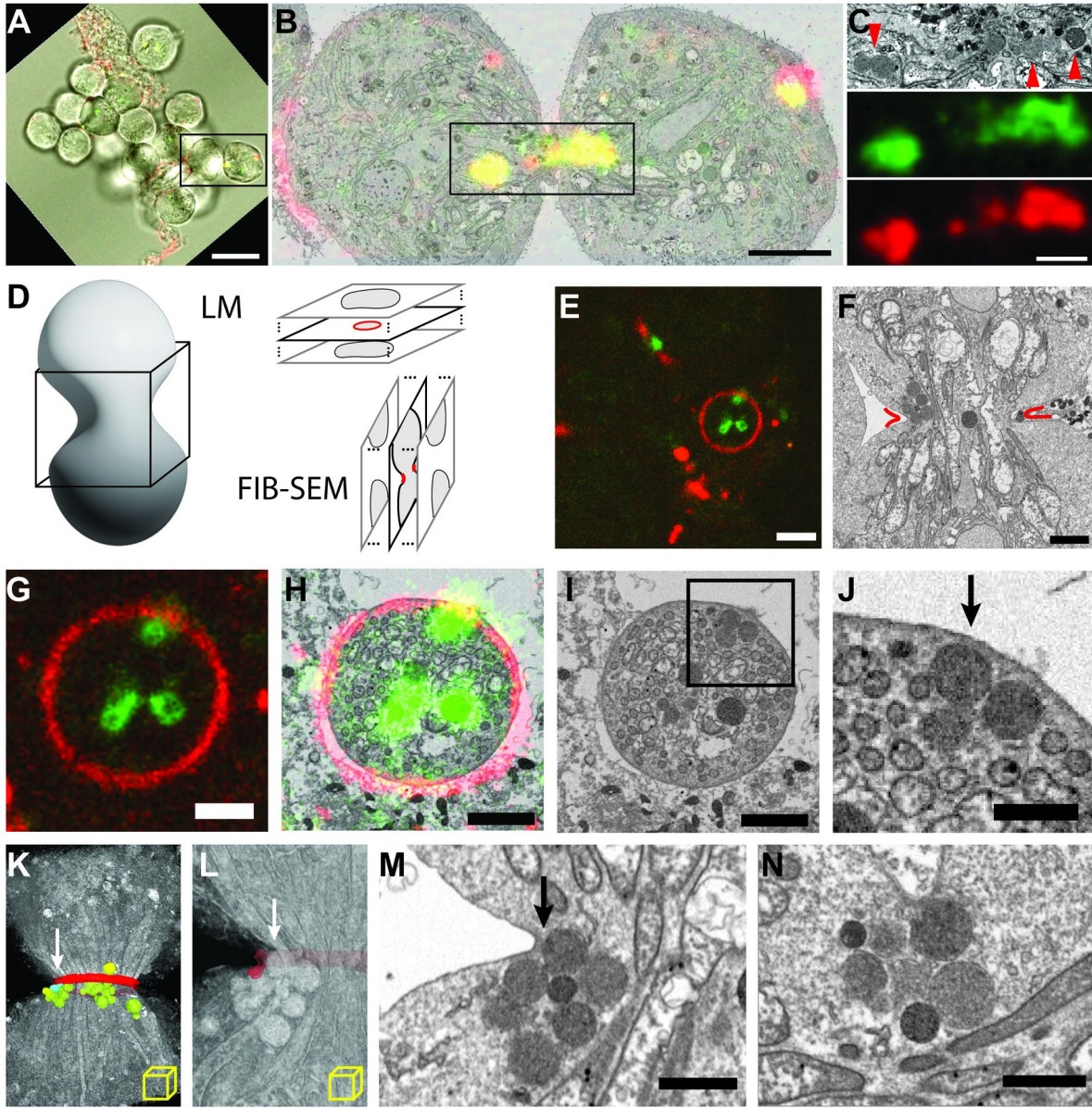

**Fig 7. CPE positive endosomes dock at the ingressing cytokinetic furrow during meiosis.** (A) Combined transmitted and fluorescence image of dissected *Drosophila* cyst; dividing cells chosen for FIB-SEM imaging are boxed. Green, eYFP-Rab7; red, PlyA2-mCherry. Scale bar: 20 μm. (B) 2D section from FIB-SEM reconstruction correlated with "z slice" from confocal. Colors as in A, saturated for ease of visualization. Scale bar: 5 μm. (C) Area boxed in B; FIB-SEM image (top), Rab7 (middle), PlyA2 (bottom). Arrowheads indicate electron-dense endosomes. Scale bar: 2 μm. (D) Volumetric imaging of dividing spermatocytes, showing orthogonal imaging planes, and expected cross sections for LM and FIB-SEM. (E) Fluorescence "z slice" capturing a cleavage furrow in-plane. Scale bar: 4 μm. (F) FIB-SEM image capturing same furrow as in E cross-section, approximated by red lines. Scale bar: 2 μm. (G–I) Correlated fluorescence image, overlay, and rotated FIB-SEM section, respectively, of furrow in-plane. (J) Area boxed in I showing endosomes docked at the furrow (arrow). Scale bar: 1 μm. (J and K) Volume rendering of FIB-SEM 3D reconstruction, with furrow segmented and false colored red, undocked endosomes green, docked endosomes blue. (L) Close-up of K, with furrow segmented translucent and endosomes unsegmented. (M) Matching FIB-SEM section in the imaging plane. Arrows in K, L, M show same endosomes docked at furrow. Scale bar: 2 μm. (N) Independent CLEM/FIB-SEM experiment revealing at higher pixel sampling another example of docked vesicle. Scale bar: 1 μm. CPE, ceramide phosphoethanolamine; LM, light microscopy.

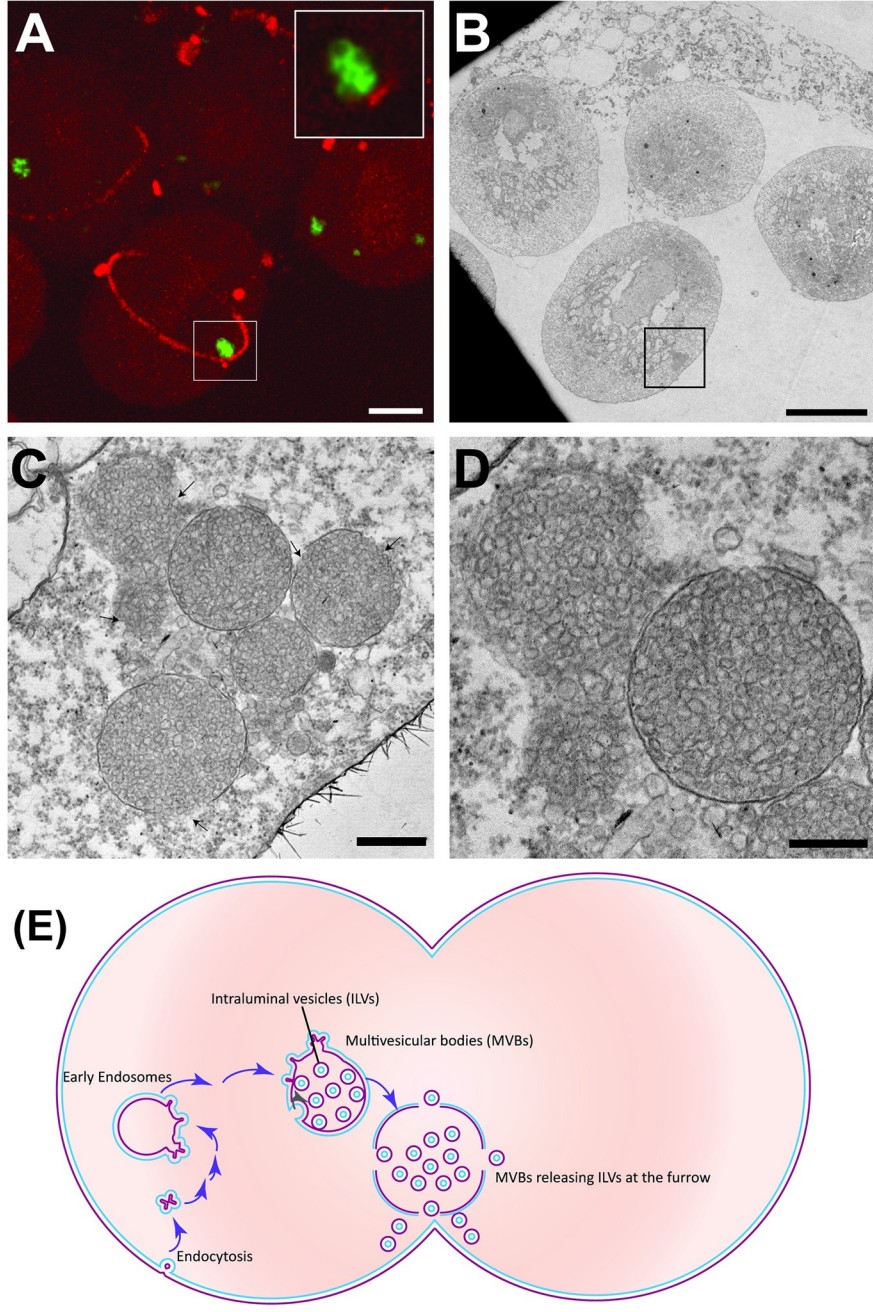

**Fig 8. MVBs release intraluminal vesicles at the cytokinetic furrow.** (A) Fluorescence maximum projection image capturing dividing cells from a dissected *Drosophila* cyst. Green, eYFP-Rab7; red, mRFP-Anillin. Inset, vesicles adjacent to a cleavage furrow. Scale bar: 10 μm. (B) Corresponding TEM section. Endosomes adjacent to the cleavage furrow are boxed. Scale bar: 10 μm. (C) Higher magnification TEM image of area boxed in B. Scale bar: 500 nm. Arrows indicate discontinuity in outer membranes of MVBs. (D) Subsequent TEM section of same area, showing make-up of vesicles at high magnification. Scale bar: 300 nm. (E) A model depicting endocytosis of CPE from plasma membrane, the maturation of these endosomes into multivesicular bodies, targeting and release of intraluminal vesicles at the cleavage furrow. CPE, ceramide phosphoethanolamine; MVB, multivesicular body.

## Discussion

Plasma membrane expansion and ingression of the cytokinetic furrow still remain one of the least understood horizons in the field of cell division [74]. We are only beginning to appreciate the involvement of lipids in structural integrity, membrane expansion, and transmission of signals during cytokinesis. PI(4,5)P$_2$ is the most studied lipid in cytokinesis and has been shown to localize to the cleavage furrow during ingression. In addition to PI(4,5)P$_2$, cholesterol, very long chain fatty acids, and sphingolipids have also been associated with cytokinesis [10,12,75]. In mitotic HeLa cells, PI(4,5)P$_2$ accumulation at the inner leaflet and subsequent recruitment of RhoGTPase (RhoA) was shown to be coordinated with sphingomyelin rich domains on the outer leaflet [10]. However, it is unknown if a similar mechanism exists in germ cells that have different characteristics and different lipid composition. While mitotic cells were shown to accumulate classical sphingolipids with long to very long acyl chains at intercellular bridges [12], male germ cells accumulated complex sphingolipids with very long/ultra-long chain polyunsaturated fatty acids (VLC-PUFAs) [17]. These differences in sphingolipid composition with distinct biophysical properties between mitotic cells and male meiotic cells point toward distinct underlying mechanisms.

In this study, we show the sphingolipid composition of the major species in *Drosophila* testis for the first time and demonstrate significant enrichment of unique CPE species containing SPH_MUFA and SPD_SFA. Unlike mammalian testis, the *Drosophila* testis lacks significant amounts of VLC-PUFA containing sphingolipids. However, due to conserved unsaturation in acyl chains, it is likely that SPH_MUFA and SPD_SFA containing CPE could substitute for VLC-PUFA functions in *Drosophila* male germ cells. Glycosphingolipids with VLC/ULC PUFAs have been shown to be required for mammalian male meiotic cytokinesis but their actual role has not been delineated [17,76,77]. It is interesting to note that CPE mimics certain physical properties of galactosylceramide (GalCer), a major sphingolipid in myelin sheath [78]. Previous studies have shown that CPE has an essential role in axonal ensheathment and cortex glial plasma membrane expansion in *Drosophila* [27,79]. It was hypothesized that the axonal ensheathment in mammals and *Drosophila* is based on similar physical process with different lipids [78]. We have previously shown that sphingomyelin, a structural analog of CPE, could rescue cortex glial membrane defects in *cpes* mutants, suggesting the head group is not as important as the tail in glial membranes. However, here we show that sphingomyelin could not rescue spermatogenesis defects in *cpes* mutants alluding to the importance of the ethanolamine head group in spermatocytes. Taken together, we could suggest that *Drosophila* CPE would have evolved to have both the properties of mammalian glycosphingolipids and sphingomyelin to promote spermatogenesis and glial ensheathment of neurons.

Phosphoinositides play critical role in cytokinesis. In particular, PI(4,5)P$_2$ was shown to be enriched at the furrows and directly binds to several proteins such as anillin, septins, and MgcRacGAP to mediate furrow stability [23,80]. Unlike mitotic cells, PI(4,5)P$_2$ in male meiotic cells does not show selective accumulation at cleavage furrows. Instead, it is uniformly distributed throughout the plasma membrane including at furrows ([61] and Fig 4C and S5, S6, and S7 Movies). Interestingly, PI(4,5)P$_2$ hydrolysis and calcium release were shown to be required for cytokinesis in *Drosophila* spermatocytes [61]. These observations suggests that in meiotic cells, PI(4,5)P2, has a broader role at the plasma membrane and at furrows to promote cytokinesis. Intriguingly, we show that the PI(3,4,5)P$_3$ binding PH domain of Steppke (tGPH) localizes to cleavage furrows indicating a potential role for PI(3,4,5)P$_3$ during male meiotic cytokinesis. However, the accumulation of PI(3,4,5)P$_3$ to the cleavage furrows occurs only in the intact cysts, i.e., 16 interconnected spermatocytes wrapped up by 2 somatic cyst cells suggesting the need for certain mechanical force. The role of PI(3,4,5)P$_3$ accumulation in the male

meiotic cleavage furrows is an open question and warrants future investigation. Notably, recently it was shown that cytohesin Steppke reduces tissue tension by inhibiting the actomyosin activity at adherens junctions in embryos [62]. Although broad accumulation of PI(4,5)P$_2$ to the plasma membranes and PI(3,4,5)P$_3$ accumulation at the cleavage furrows were not altered in *cpes* mutants, their specific localization to membrane microdomains and their inability to signal robustly cannot be ruled out. Sphingadienes with $\Delta^{4,6}$ conjugated double bonds were previously identified in *Drosophila*, *Manduca sexta*, and *B. mori* [81–83], although their enrichment in the testis was not known until now. However, normal sphingolipid synthesis and their degradation are shown to be required for spermatogenesis [84,85]. The sphingadiene sphingolipids were shown to have inhibitory effects on AKT-dependent signaling [86] and down-regulate *Wnt* signaling via a PP2A/Akt/GSK3β pathway [87]. Thus, accumulation of sphingadiene with saturated fatty acid containing sphingolipids in the testis could have a role in regulation of PI(4,5)P$_2$/PI(3,4,5)P$_3$ mediated signaling in vivo. Apart from CPE_SPD_SFA, the *Drosophila* testis is also enriched in CPE_SPH_MUFA suggesting they could mediate distinct functions. Lipids with unsaturated fatty acids mediate a number of biological functions [88]. It is widely known that phospholipids with unsaturated fatty acids promote membrane fluidity and elasticity [89,90]. Sphingolipids with unsaturated fatty acids could thus provide unique biophysical properties necessary for the membrane curvature requirements during meiotic cytokinesis. DES-1 codes for dihydroceramide desaturase enzyme in the de novo biosynthetic pathway in *Drosophila*. *des-1* mutants (also called *infertile crescent*, *ifc*) were described before the gene product was identified as a homolog of dihydroceramide desaturase. The primary spermatocytes of the *des-1* mutants undergo degeneration without initiating chromosome condensation at the beginning of meiosis [91]. A P-element insertion allele showed that DES-1 colocalized with mitochondria and was intimately associated with the central spindle of dividing meiotic spermatocytes. Its deficiency leads to failure of central spindle assembly in these dividing cells leading to several phenotypes including cytokinetic defects. It was proposed that DES-1 could be part of an anchoring mechanism that linked membrane bound cellular compartments to components of the cytoskeleton [84]. Considering that DES-1 is a dihydroceramide desaturase homolog, it would be worth revisiting the mutant phenotype to evaluate if it too contributes to membrane assembly at the cytokinetic furrow due to lack of CPEs.

In correlative live imaging and electron microscopy studies, we find that MVBs target CPE laden vesicles to the cleavage furrow. MVBs are by definition spherical organelles that are typically 250 to 1,000 nm in diameter with a single outer membrane that enclose a variable number of smaller spherical vesicles within. Originally, they were believed to be involved in neurosecretion because of their close association with the Golgi stacks [92]. Further work established a primary role in endocytosis and subsequent degradation of proteins [93–95]. Other works have highlighted the important role of MVBs in recycling function [96]. The last decade or so has seen the emergence of a role for MVBs in autophagy and secretion via the exosomes [97,98]. Our work has surprisingly unraveled a possible new function for MVBs in transporting the endocytosed vesicles rich in lipids to the vicinity of expanding membranes. Interestingly, CPE is not required for localization of MVBs to the cleavage furrow (S21 Movie), instead, our data suggests that CPE loaded MVBs are required in the vicinity of expanding membranes to mediate cytokinesis. Consistent with this possibility, our genetic analysis showed that ablation of Rab11 function prevents cleavage furrow-specific localization of CPE positive MVBs which in turn could partly explain increased cytokinetic defects in Rab11 mutants or in Rab11 DN expressing genetic background. Rab11 and Rab35 are small GTPases whose roles in membrane recycling during late steps of cytokinesis have been well established [22–24,70,99]. In mammalian system, it was shown that FIP3 (Rab11 family interacting

proteins), an effector of Rab11 binds simultaneously to Rab11 and Arf6 (small GTPase) through distinct sites [100]. Rab11-FIP3 endosomes move in and out of intercellular bridge with the help of microtubule-based motors kinesins (KIF5B) and dynein that are regulated by Arf6 and KIF5B-dynein scaffolding protein JIP4 [101]. In *Drosophila*, Nuclear fallout, a FIP3 homologue was shown to bind Rab11 and regulate actin cytoskeleton remodeling during early furrow formation in embryos [99]. Membrane trafficking transport protein particle (TRAPP) II complex component brunelleschi (bru) was shown to genetically interact with Rab11 and is required for cleavage furrow localization of Rab11 in *Drosophila* spermatocytes [102]. Phosphatidylinositol 4 phosphate (PI (4)P) effector protein Golgi phosphoprotein 3 (GOLPH3) was shown to be essential for contractile ring formation and Rab11 localization to cleavage furrow in *Drosophila* spermatocytes [26]. Further, recruitment of Rab11 to the cleavage site also depends on wild-type functions of exocyst complex subunits Exo84 and sec8 [21]. Taken together, these studies suggest that Rab11 localization to cleavage furrow is regulated by multiple proteins including microtubule motors, transport proteins, Golgi components, and exocyst components. Therefore, it is likely that Rab11 complexed with yet other unknown proteins could mediate trafficking of CPE positive MVBs to the cleavage furrow, it could mediate docking via interaction with exocyst components and mediate fusion via SNAREs on target membranes [23]. Although Rab35 mutants show cytokinetic defects, we were unable to localize Rab35 to the endosomal compartment in fixed spermatocytes. We show Rab35 is not required for localization of CPE positive MVB to the cleavage furrow; however, we cannot rule out if Rab35 plays a role after CPE localization to furrow membranes. How Rab35 mediates cytokinesis in spermatocytes remains an open question. Similarly, absence of Rab7 does not prevent endocytosis and targeting of CPE positive MVBs to cleavage furrow. However, presence of cytokinetic defects in Rab7 mutants indicates certain dynamics of CPE positive MVBs could be compromised. Future studies should throw light on the role of this CPE positive MVB pathway in membrane and protein accumulation in newly synthesized membranes as in the case of meiotic cytokinesis in the testis. The human genome encodes more than 60 Rab proteins and several of them including Rab1, Rab8, Rab10, Rab14, Rab21, Rab24, and Rab35 were shown to localize to intercellular bridges [23]. Currently, it is unknown if these endosomes are also involved in delivery of specific plasma membrane-derived lipids to the cleavage furrows. Our study shows that there is an increased localization of CPE to the endosomal compartments in spermatocytes. The CPE containing MVBs are selectively docked on the ingressing membranes at the cytokinetic furrow and release intraluminal vesicles, suggesting delivery of CPE rich membranes to the growing furrows. Future investigations focusing on these CPE-enriched endosomes and their dynamics during male meiotic cytokinesis, spermatid polarity, and individualization will provide novel insights into membrane expansion and or their role as a signaling center to mediate spermatogenesis.

## Materials and methods

### Fly stocks

*tubulin*-Gal4 (BDSC#5138), UAS-PLCδ-PH-EGFP (BDSC#39693), tGPH (BDSC#8163), Sqh-GFP.RLC (BDSC#57145), Sqh-mCherry.M (BDSC#59024), p(Ubi-p63E-Feo-mCherry)3 (BDSC#59277), *bam*-Gal4 [51], *nos*-Gal4 (BDSC#4937), *chif*-Gal4 (BDSC#13134), C587-Gal4 (BDSC#67747), EYFP-Rab4 (BDSC#62542), EYFP-Rab6 (BDSC#62544), EYFP-Rab7 (BDSC#62545), EYFP-Rab11(BDSC#62549), UAS-EGFP (BDSC#5430), tub-Gal80ts (BDSC#7108), UASP-Rab11.S25N (BDSC#23261), UAS-mRFP-Anillin (BDSC#52220), rab7 mutant [69], Rab35 mutant [71], genomic CPES_V5tag [27], *cpes* mutants [27], UAS-hSMS1

[27], dcert1[43], UAS CDase [44], UAS-CPES active site mutants, UAS-Aedes CPES, UAS--Bombyx CPES (this study).

## Fly husbandry

All *Drosophila* stocks were raised on standard fly food and maintained 25˚C unless otherwise mentioned. Due to increased lethality, *cpes* mutants were raised at 18 to 21˚C. Certain genetic backgrounds increased pupal death in *cpes* mutants such as overexpression of UAS CDase, UAS-PLCδ-PH-EGFP, *cpes*; *dcert*[1] double mutants. In such cases, we have separated third instar larvae of *cpes* mutants and allowed them to develop into pupae. The testes were dissected before the death of pupae and analyzed.

## Cloning of UAS-*Aedes aegypti* CPES and UAS-*Bombyx Mori* CPES

The coding sequence for *A. aegypti* CPES (XM_021851518.1) and *B. mori* CPES (XM_004923146.3) were codon optimized and synthesized by GenScript and subcloned into pUAST vector. The clones were sent to BestGene for embryo injection service. The transgenic flies were balanced and crossed with Gal4 drivers to conduct rescue experiments.

## Cloning of CPES active site mutant

Two conserved aspartates in CPES active site were mutated to alanine using PCR-based site directed mutagenesis method as described by [103]. PCR was performed using pBluescript SK + CPES_V5 clone as template with appropriate sense and antisense primers (S1 Table) and Phusion polymerase (NEB). The PCR product was digested with DpnI to remove templates followed by transformation into DH5α competent cells. Plasmids were isolated from colonies grown in LB media and screened by restriction digestion with appropriate restriction enzymes (S1 Table) and sequencing. The pBluescript SK+ CPES DD-AA clone was used as template to PCR amplify the insert using appropriate primers (S1 Table), restriction digested and cloned into pUAST vector. The clones were confirmed with restriction digestion followed by DNA sequencing.

## Cloning, expression, and purification of PlyA2mCherry

Mushroom-derived aegerolysin PlyA2 was cloned, expressed, and purified as described by [104]. Briefly, the coding sequence of PlyA2 (accession number AB777517) fused to mCherry at the C-terminus was synthesized by GenScript. The linker amino acid sequence between PlyA2 and mCherry was VDGTAGPGSIAT. This clone was used as template to PCR amplify using appropriate forward and reverse primers (S1 Table) and cloned into p24a vector at appropriate restriction sites (S1 Table). The clones were transformed into *Escherichia coli* strain BL-21 (DE3) and single colony was inoculated in to 5 ml of LB broth containing 50 μg/ml kanamycin and grown overnight in orbital shaker (200 rpm) at 37˚C, and 2 ml of overnight culture was inoculated into 250 ml of fresh LB broth containing kanamycin and grown at 37˚C until the optical density reached to 0.6. Subsequently, the culture was incubated in cold (4˚C) for 2 h, followed by induction with 0.3 mM isopropyl-B-D-thiogalactopyranoside, and grown at 25˚C in orbital shaker (200 rpm) overnight. The cells were harvested by centrifugation at 5,000g for 10 min at 4˚C, and the pellet was resuspended in binding buffer (20 mM sodium phosphate buffer (pH 7.4), 500 mM NaCl, 20 mM imidazole) containing EDTA free protease inhibitor cocktail (Sigma, P8340). The cell suspension (50 ml) was sonicated at 4˚C (Cole-Par-mer ultrasonic processor model CP130,) with microprobe at an amplitude of 30 and on and off cycles of 10 s each for about 30 min to 1 h. The lysate was centrifuged at 10,000g for 10 min

to remove insoluble proteins and debris. To the supernatant, 1 ml of 50% Ni-NTA agarose (Thermo Scientific) was added and incubated on end-to-end rotor for 3 h and the beads were collected on a column and washed with 30 ml of binding buffer and eluted with binding buffer containing 200 mM imidazole. The fractions were dialyzed in 20 mM sodium phosphate buffer 3 times (1 L each), and the final protein was concentrated with Amicon centriprep filter devices for volumes up to 15 ml. The protein concentration was determined using Bradford assay. The proteins were resuspended in 20% glycerol, aliquoted (100 µl), and stored at −80˚C for future use.

## Testis squash preparation, phase contrast microscopy, and immunofluorescence microscopy

*Drosophila* testis squash preparations for phase contrast microscopy and immunostaining were performed as described by [29]. Briefly, young male (0 to 2 days old) testes were dissected (5 pairs) in phosphate-buffered saline (PBS, 130 mM NaCl, 7 mM Na2HPO4, 3 mM H2PO4). The testes were torn open on a slide at appropriate location with a pair of black anodized steel needles (tip diameter 0.0175 mm) and a cover glass was gently placed without air bubbles. The slides were directly observed under phase contrast microscopy or snap frozen in liquid nitrogen for immunostaining. After freezing, the cover glass was removed from the slide with razor blade and immersed in cold 99.5% ethanol and incubated in −20˚C for 10 min. The slides were then fixed with 4% paraformaldehyde (PFA) in PBS containing 0.1% Triton-X-100 (PBST) for 15 min. Subsequently, the slides were washed with PBS and the area of squashed tissue was circled with a hydrophobic barrier pen, permeabilized with PBST (30 min), and blocked with 5% normal goat serum in PBST (1 h at RT). Squashed tissues were incubated with primary (1 µg in 100 µl) and secondary antibody (1:100 dilution) (8 to 12 h each at 4˚C in a moist chamber). Slides were washed with PBST (3× 10 min each) after each antibody treatment, finally incubated with DAPI solution (1:2,000 dilution from 5 mg/ml stock, Molecular Probes) and mounted with Vectashield H1000 mounting media.

## Quantification of cytokinetic defects

Five pairs of testes were dissected for each experiment, live testis squashes were prepared, and round spermatids were observed using 40× Ph2 Plan-Neofluor objectives on Axioplan 2 microscope as described above. All round spermatid cyst in the sample were imaged and their cell number was counted manually (250 to 500) for each genotype. Percentage of round spermatids with 2, 3, or 4 nuclei were calculated by dividing the total spermatid number including spermatids with 1, 2, 3, or 4 nuclei. More than 5 independent experiments were carried out for each genotype.

## Male fertility assay

Male fertility assay was performed by setting up individual crosses with 1 male and 3 wild-type females. Each replicate involved 30 males and 3 independent replicates (total of 90 males) were performed for each experiment.

## Immunostaining and confocal imaging

*Drosophila* tissues were dissected and immunostained as described previously [27]. Briefly, dissected testes were fixed with 4% PFA in 0.3% PBST for 1 h at RT, washed (3× 10 min each), permeabilized in 0.3% PBST (1 h), blocked with 5% NGS (1 h), and incubated with primary antibody (1:100 dilution or 1 µg in 100 µl of 5% NGS in PBST) overnight (8 to 12 h). Testes

were washed with 0.3% PBST (3× 10 min each) and incubated with secondary antibody (1:100 dilution in 5% NGS in PBST) overnight (8 to 12 h). Testes were washed with PBST (3× 10 min each), stained with DAPI (1:2,000 dilution of 5 mg/ml solution in PBST) for 15 min, and mounted with Vectashield H1000 mounting medium. The slides were imaged on ZEISS confocal laser scanning microscope LSM 780 or LSM880.

## Live cell imaging of meiotic cysts

About 15 to 20 pairs of young (0 to 2 days old) *Drosophila* testes were dissected in 600 μl of insect cell culture media (M3 media supplemented with heat inactivated FBS, Pen/Strep) in a glass well plate. Testes were transferred to a fresh well with 400 μl of media and torn open with a pair of black anodized steel needles (tip diameter 0.01 mm) to release the cysts into media. This was repeated until all the testes were torn open. They were gently agitated so that all the cysts were released into the media followed by removal of empty testis or debris from the media. The cysts were gently washed with fresh 400 μl media (2×), then transferred via pipetting into the middle of 35-mm cover glass bottom dish (Poly D-Lysine coated) and fresh media (max. 200 μl) was gently added to the cleft formed between cover glass and the dish. The cysts tend to move while handling but remain settled on the cover glass and ideally remain in position while imaging. The intact cysts and cysts that have lost their cyst cells could be distinguished in bright field view. The cysts that are beginning to undergo meiosis could be identified in bright field view by looking at cell morphology and differences in nucleus and spindle appearance. Cells beginning to undergo meiosis could also be identified by Feo-mCherry that is localized to nucleus in interphase but translocate to cytosol during prophase to anaphase. Prophase to anaphase cysts were identified using 25× water objective and subsequently imaged using 63× water objective on a Leica Andor spinning disk confocal microscope. The time lapse imaging was done at 2 min intervals with the z stacks (about 80 slices, 0.5 μm per slice) at each time point for 60 to 90 min for completion of meiotic I cytokinesis. Laser power and exposure times were preferably kept low to prevent bleaching and aberrant effects on cytokinesis due to generation of free radicals.

## Sample preparation for correlative light and electron microscopy (CLEM)

About 10 to 15 pairs of testes were dissected from newly eclosed male flies in M3 or S2 insect cell culture media. Testes were transferred to fresh M3 media (400 μl) in a glass spot plate (PYREX 722085) and torn open with a pair of black anodized steel needles and gently agitated to release the cysts into media. Subsequently, testis debris were removed from the well. The cysts were gently washed with fresh 400 μl media (2×) and incubated with PlyA2-mCherry protein (10 μg/ml) in 400 μl of M3 media for 1 h, on a rocker. The cysts were washed with fresh media without PlyA2-mCherry (1× with 400 μl media). About 5 to 10 cysts were transferred to cover glass bottom 96-well plate (Cellvis, Cat # P96-0-N) containing 100 μl of M3 media. The cysts in each well were observed under 25× water objective on an Andor spinning disk microscope using bright field and/or fluorescence. Cells undergoing division were identified by cell morphology changes such as cell rounding, nuclear membrane loss, microtubule appearance, cell elongation, and ingressing cleavage furrows. When spermatocytes at the appropriate stage of interest were identified, 100 μl of 8% PFA in M3 media was added to the well (4% PFA final concentration) and gently mixed with pipetting. The 96-well plate cover was removed after adding PFA to avoid its effect on other wells. Location of the well could be identified with the help of a torch light and/or illuminating the well with bright field or fluorescent light. The process was repeated until at least 5 to 10 cysts were identified and fixed. The sample in fixative could be stored up to 2 days at 4°C in the 96-well plate. The cysts in the

96-well plates were gently agitated with pipetting and transferred to a glass spot plate (the pipet tip was cut and conditioned with the media to avoid sticking of cysts to tip walls). At this stage, cysts were pooled from different 96 plate wells into a single glass well plate. The cysts were centered by gently flushing with pipetting. They were then washed with 1XPBS (3× 400 μl). Following washing, the cysts were pipetted into the center of a gridded covered glass bottom plate (MatTek, Part No. P35G-1.5-14-C-Grid). After the cysts settled on the coverslip, most of the PBS was removed by gentle pipetting. Subsequently, 4% low melting agarose (Invitrogen, REF 16520–100) (in 1xPBS) that was maintained at 80˚C is cooled for 1 to 2 min and gently poured from the top, care must be taken to minimize movement of cysts. After the agarose had solidified, the cysts that were settled and embedded in the agarose were imaged on the Andor spinning disk confocal microscope using 25× and 63× water objectives. At this time, the cells were imaged at high magnification by transmitted light to record the location of regions of interest (ROI) with respect to the alphanumeric grid of the glass-bottom dish. Images were also taken at lower magnification to record the locations of other cells in the dish, which would later serve as landmarks in the correlative light-electron microscopy (CLEM) pipeline; images of the ROI at various points in this pipeline are shown in S11 Fig.

The disc of solidified agarose containing the cysts was carefully removed from the glass-bottom dish. Not only are the cells embedded in situ typically within 10 microns of the bottom agarose surface, but also the alphanumeric gridded pattern is transferred to the agarose pad in relief, and both are visible in a dissection microscope. The agarose pad was rinsed with 0.1 M sodium cacodylate buffer (11652, Electron Microscopy Sciences) several times. It was then incubated in a solution containing 2% osmium tetroxide (19150, Electron Microscopy Sciences) and 1.5% potassium ferricyanide (20150, Electron Microscopy Sciences) in 0.1 M sodium cacodylate buffer for 1 h at room temperature. Afterwards, it was thoroughly washed with water for a minimum of 30 min. The sample was then incubated in 1% aqueous uranyl acetate (22400, Electron Microscopy Sciences) for 30 min followed by extensive water washes. This was followed by standard EM processing steps—the sample was dehydrated using a series of increasing ethanol concentrations ending in 100% ethanol and then washed with 100% propylene oxide. The sample was infiltrated with Epon (Embed 812) resin using increasing concentrations of hard resin formulation in propylene oxide, and finally embedment in a polypropylene dish, with curing in an oven at 60˚C for 2 days. Once the resin was fully cured, it was separated from the dish and examined using a dissecting microscope. The heavy metal-stained cysts were dark and easily identified; the overall pattern of cysts was unchanged from previous steps (S11 Fig). ROIs for CLEM were identified and approximately 1 mm$^2$ areas around each cyst were cut out from the larger disc of resin using a saw and razor blades. These were then re-embedded onto blank blocks for ultramicrotomy. The blocks were sectioned using an ultramicrotome (Powertome, RMC) until cysts were exposed at the resin surface as revealed by Toluidine Blue stain, taking care that the actual cleavage furrow itself was just below the surface and not sectioned. This positions the feature within approximately 10 microns of the top surface and therefore amenable to FIB-SEM imaging.

### FIB-SEM, STEM, and TEM imaging

FIB-SEM imaging was done largely as previously described [105]. Briefly, the specimen was cut to approximately 2 mm height and mounted on an SEM stub using silver. The specimen was introduced into a Zeiss CrossBeam 550 (Zeiss), ROI located by SEM imaging, and protected by a patterned platinum and carbon pad. A trench was FIB milled until the profiles of both dividing cells were revealed but stopped before the cleavage furrow was milled away. Images were acquired at 3 nm or 5 nm pixel sampling in XY and step size of 9 or 15 nm in Z,

respectively, in an automated FIB-mill-SEM-image cycle with SEM operated at 1.5 kV, 1.2 nA; FIB milling at 30 kV, 1.5 nA, and back-scatter detector grid voltage at 900 V. The image stack was registered, contrast inverted, and binned by 3 in the imaging plane to produce an isotropic $9 \text{ nm}^3$ or $15 \text{ nm}^3$ image volume reconstruction. For TEM imaging, the cysts were exposed using the same approach, but here, 60-nm serial sections were cut and collected onto TEM grids. STEM imaging was executed at a Zeiss Gemini II SEM 450 at 30 kV landing energy, and the TEM imaging was done on a Hitachi 1050 operated at 80 kV. In both cases, the grids were not post stained; however, approximately 4-nm carbon coat was applied using a sputter coater (Leica) before imaging.

### Image correlation and visualization

There were 2 stages of correlation in these experiments. The correlation for relocation, i.e., imaging an ROI by LM and then locating the same ROI for EM, was performed by careful sample preparation and appropriate imaging at various stages (S11 Fig). Correlation for registration, i.e., aligning 2 image volumes to identify cellular features of interest, was done after LM and EM images were acquired. 3D correlation was done using eC-CLEM [106]. The confocal stack and FIB-SEM image stacks were imported, and multiple fiducials were placed at distributed locations on the plasma membrane of both dividing cells in the PlyA2-mCherry + eYFP-Rab7 expressing cysts. The plasma membrane was visualized easily in the FIB-SEM images, and more approximately in the lower resolution LM data, using the PlyA2-mCherry signal that localizes strongly at the cell membrane. The green channel (Rab7) was not used for registration. The skew transformed LM fluorescence was overlaid on the original FIB-SEM data and examined with the green channel added back, upon which the identity of the PlyA2 + Rab7 double positive features in the furrow were revealed (S17 Movie). In a second experiment using mRFP-Anillin: eYFP-Rab7 cysts (Fig 7E), the correlation in ecCLEM was performed using fiducial points on the furrow, as located by the "neck" in the FIB-SEM image volume (Fig 7F), and the mRFP-Anillin staining in the LM images (Fig 7G). As before, the Rab7 signal was not used for correlation. In both cases, the registration was limited by the relatively lower resolution of the LM image stack but was sufficient to consistently correlate large vesicles in the FIB-SEM data with Rab7-positive signals in the LM data. For TEM imaging, sequential sections were imaged and inspected manually until the features captured in the LM started to appear in the EM sections. The images were overlaid but not computationally correlated, as the identity of the features was readily apparent in the 2D EM image. The FIB-SEM image volumes were visualized, and volume rendered using Arivis (Arivis), with features of interest such as furrow, docked and undocked vesicles segmented either in 3DSlicer (www.slicer.org) or Arivis.

### Sphingolipid estimation in fly tissues using supercritical fluid chromatography coupled to mass spectrometry (SFC/MS/MS)

Sphingolipid estimation using supercritical fluid chromatography coupled to mass spectrometry (SFC/MS/MS) was performed as described previously [27]. Lipids were extracted from heads, ovary, and testis samples as described [107,108]. About 400 fly heads, about 250 pairs of ovaries and about 250 pairs of testes were used for each biological replicate. Three independent replicates were taken for each sample. For Figs 1 and 2, only testes samples were used, whereas for S5 Fig, whole male organ (testis, accessory glands, collecting ducts, and ejaculation bulb) were used for lipid extraction. Internal standards (ISTD) added to the tissues before extraction included 500 pmol (20 µl) of Cer/Sph Mixture I (Avanti Polar Lipids LM-6002) and 1 nmol of C12 Sphingosyl PE (C17:1/C12:0) (Avanti Polar Lipids). Lipids from all samples were

normalized to carbon content (100 μg/ml) and 100 μg/ml sample was diluted 10 times before injection for SFC. We have used large-fragment MRM method for quantitation of amount per 100 μg of carbon.

## Total RNA sequencing

Three independent biological replicates were performed for each sample and about 200 pairs of $w^{1118}$ (WT 1–3), *cpes* mutant (KO 1–3), and *bam*-Gal4>UAS-CPES rescued (RES 1–3) fly testes were dissected for each replicate. The testes were dissected in S2 cell culture media, washed with PBS, and RNA was extracted. Total RNA was extracted using Trizol method followed by DNAase I treatment and purification using RNA Clean and Concentrator-5 kit (Zymoresearch). A total of 2 μg of RNA was submitted to Novogene for total RNA seq. The RNA seq data was further analyzed for differential gene expression and pathway analysis using GSEA. The gene sets specific for *Drosophila* were downloaded from http://www.bioinformatics.org/go2msig/ and http://ge-lab.org/gskb/. The gene sets specific for cell types in *Drosophila* testis are described in Witt and colleagues, 2019 [45].

## Supporting information

**S1 Fig. Comparison of hexosylceramide subspecies in *Drosophila* tissues.** (A) Hexosylceramides composed of sphingosine base linked to saturated fatty acid (HexCer_SPH-SFA) and its subspecies are shown as picomols (pmol) in 100 micrograms of carbon. WTHM, WTHF, WTOY, and WTTS. (B) Hexosylceramide subspecies composed of sphingosine base linked to monounsaturated fatty acid (HexCer_SPH_MUFA). (C) Hexosylceramide subspecies composed of sphingosine base linked to polyunsaturated fatty acid (HexCer_SPH_PUFA). (D) Hexosylceramide subspecies composed of sphingadiene base linked to saturated fatty acid (HexCer_SPD_SFA). (E) Cartoon showing the representative chemical structure of 3 major and 1 minor HexCer subspecies including HexCer_SPH_SFA, HexCer_SPH_MUFA, HexCer_SPD_SFA, and HexCer_SPH_PUFA, respectively. Statistical significance was calculated using mean, SD, and N in Prism 8. The 2-way ANOVA multiple comparison was used to calculate *p*-values where **** $p \leq 0.0001$, *** $p \leq 0.001$, ** $p \leq 0.01$, * $p \leq 0.05$, and ns $p > 0.05$. Three independent biological replicates were taken for each sample and lipids were extracted from 400 heads or 250 pairs of ovary or testis for each biological replicate. The data underlying the graphs shown in this figure can be found in S1 Data. SD, standard deviation; WTHF, wild-type heads from female; WTHM, wild-type heads from male; WTOY, wild-type ovary; WTTS, wild-type testis.
(PDF)

**S2 Fig. CPES enzymatic activity is required for spermatogenesis and accumulation of ceramide is not responsible for cpes mutant phenotypes.** (A, C, and E) Live testis squash preparation followed by phase contrast microscopy of round spermatids; (B, D, and F) are immunofluorescence images of early elongating spermatids from samples corresponding to A, C, and E, respectively. Immunostaining was performed with beta tubulin primary antibody and Alexa Fluor 568 conjugated secondary antibody (red). Nuclei are stained with DAPI (blue) (A and B), the UAS-CPES active site mutant ($GX_3DX_3D$ to $GX_3AX_3A$) was expressed using bam-Gal4 in cpes mutant background. (C and D), cpes; dcert[1] double mutants. (E and F) The UAS CDase (neutral ceramidase) was expressed using bam-Gal4 driver in cpes mutant background.
(PDF)

**S3 Fig. Analysis of cell type-specific transcriptomic signatures in cpes mutants.** (A) GSEA using top 50 cell type-specific expressed gene sets and compared between cpes versus WT and bam-Gal4>UAS CPES rescue (Rescue). Red bars indicate gene sets enriched in cpes mutants and blue bars indicate gene sets enriched in WT and Rescue. (B) Enrichment plots for each of the cell type show the distribution of individual genes in each set (vertical lines on red to blue parallel bar). Genes with positive ES indicate that they are enriched in cpes mutants whereas negative ES indicate that they are enriched in WT and Rescue. (C) Heatmap comparison of cell type-specific enriched gene sets in WT ($w^{1118}$), cpes, and Rescue (bam-Gal4>UAS CPES rescue). Red indicates gene sets enriched in cpes mutants and blue indicate gene sets enriched in WT and Rescue. The data underlying the graphs shown in this figure can be found in S2 Data. ES, enrichment score; GSEA, Gene Set Enrichment Analysis; WT, wild type. (PDF)

**S4 Fig. Cell type-specific expression pattern of various UAS and Gal4 constructs in *Drosophila* testis.** (A-) Whole mount *Drosophila* testis immunostained with anti-GFP antibody (green), DNA stained with DAPI (blue). (A) nanos-Gal4 (nos) drives the expression of pUASP-alpha-Tubulin-GFP, (B) bam-Gal4 drives the expression of pUASP-alpha-Tubulin-GFP, (C) chiffon-Gal4 drives the expression of alpha-Tubulin-GFP, (D) C587-Gal4 drives the expression of alpha-Tubulin-GFP, (E) nanos-Gal4 (nos) drives the expression of pUAS-T-EGFP, (F) bam-Gal4 drives the expression of pUAST-EGFP, (G) chiffon-Gal4 drives the expression of pUAST-EGFP, (H) C587-Gal4 drives the expression of alpha-Tubulin-GFP, (I) nanos-Gal4 (nos) drives the expression of pUAST-PLCδ-PH-EGFP, (J) bam-Gal4 drives the expression of pUAST-PLCδ-PH-EGFP, (K) chiffon-Gal4 drives the expression of pUAST-PLCδ-PH-EGFP, (L–N) verification of various nanos-Gal4 drivers and their expression pattern. BDSC#4937 (used in this study) and BDSC#64227 specifically express transgene in GSC and early spermatogonia compared to BDSC#32563 that expresses more broadly. (PDF)

**S5 Fig. Insect CPE synthases but not sphingomyelin synthases rescue spermatogenesis defects.** (A–C) Phase contrast microscopy of round spermatids where UAS hSMS1 (A), UAS Aedes CPES (B), and UAS Bombyx CPES (C) were ubiquitously expressed in cpes mutant background using tubulin Gal4. The scale bar is equivalent to 10 μm. (D) Measurement of cytokinetic defects in UAS-hSMS1, UAS-*Aedes aegypti*, and UAS-*Bombyx mori* CPES rescue testes; 4 independent experiments for UAS-hSMS1 (round spermatid count, $n = 500$), 2 independent experiments for UAS-*Aedes aegypti* (round spermatid count, $n = 1,100$), and 4 independent experiments for UAS-Bombyx CPES (round spermatid count, $n = 1,900$). (E–G) Confocal images of elongating spermatids immunostained with beta tubulin primary antibody and Alexa Fluor 568 secondary antibody (red) and DAPI for DNA (blue). UAS hSMS1 (E), UAS Aedes CPES (F), and UAS Bombyx CPES (G). (H–K) Sphingolipid analysis of lipids extracted from testis of wild type, cpes mutant, and various rescues expressed in germ cells using vasa-Gal4. In contrast to Figs 1 and 2, here we used whole male organ (testis, accessory glands, collecting ducts, and ejaculatory bulb) for lipid extraction. (H) The amounts of CPE and its subspecies in wild type, cpes mutant, and various rescues are shown. (I) Ceramide and its subspecies are shown in picomoles per 100 μg of carbon. (J) Hexosylceramide subspecies amounts are shown in bar diagram. (K) Amount of SM and its subspecies are shown as picomoles per 100 μg of carbon. Fluorescent blue green part of the bar represents SM species with net 2 double bonds in them. (L) TLC of lipids extracted from 50 pairs of testes from each genotype. The position of SM on TLC is indicated by an arrow. Original uncropped TLC image can be found in S1 Raw Images. DHSPH, SPH, SPD SFA, MUFA and PUFA. Statistical significance was calculated using mean, SD, and N in Prism 8. The 2-way ANOVA multiple

comparison was used to calculate $p$-values where **** $p \leq 0.0001$, *** $p \leq 0.001$, ** $p \leq 0.01$, * $p \leq 0.05$, and ns $p > 0.05$. Three independent biological replicates were taken for each sample and lipids were extracted from 250 male organs for each biological replicate. The data underlying the graphs shown in this figure can be found in S3 Data. DHSPH, dihydrosphingosine; MUFA, monounsaturated fatty acid; PUFA, polyunsaturated fatty acid; SD, standard deviation; SFA, saturated fatty acid; SM, sphingomyelin; SPD, sphingadiene; SPH, sphingosine; TLC, thin layer chromatography.
(PDF)

**S6 Fig. GSEA pathway analysis indicates enrichment of spindle elongation and spindle elongation processes.** (A) GSEA pathway analysis of RNA seq data from w[1118], cpes mutant, and bam-Gal4>UAS-CPES rescue testis. Three independent biological replicates were taken for each sample and RNA was extracted from 200 pairs of testes for each biological replicate. (B) Enrichment plot showing genes involved in spindle elongation are positively correlated with cpes mutants compared to w[1118] and bam-Gal4>UAS-CPES rescue. (C and D) Heatmap shows spindle elongation genes are enriched in cpes mutants. (E) Enrichment plot showing genes involved in spindle organization are positively correlated with cpes mutants compared to w[1118] and rescue. (F and G) Heatmaps show top 100 enriched genes in cpes mutants. (H) Heatmap shows the top 50 genes enriched in wild type and rescue. The data underlying the graphs shown in this figure can be found in S2 Data.
(PDF)

**S7 Fig. Analysis of spindle microtubules and chromosomal DNA in testis squash preparations.** Microtubules are immunostained with alpha tubulin antibody and chromosomal DNA with DAPI. Confocal images of wild-type spermatocytes in metaphase (A–C), anaphase (D), early telophase (E), and late telophase (F). Confocal images of cpes mutant spermatocytes in metaphase (G–I), anaphase (J), early telophase (K), and late telophase (L).
(PDF)

**S8 Fig. Cytological analysis of meiotic cytokinesis defects in Rab7 mutant, Rab11 DN, and Rab35 mutants.** (A) Measurement of cytokinetic defects in testes squash preparations of wild-type samples, 7 independent experiments, and total round spermatid count of about ($n = 6,000$). Fraction of round spermatids showing nucleus to mitochondria ratio 2:1 (2 nucleus), 3:1 (3 nucleus), 4:1 (4 nucleus), and 4 irregularly sized nucleus with 1 mitochondrion are depicted. (B) Cytokinesis defects in Rab11 dominant negative expressing gonads at fourth day, 5 independent experiments, and total round spermatid count of about ($n = 1,500$). (C) Cytokinetic defects in Rab35 null mutants, testis from 1–2-day-old adult males were used for analysis, 4 independent experiments, and a total of round spermatid count equals to 1,250. (D) Western blotting of protein extracts from control and Rab7 mutants using Rab7 and tubulin (loading control) antibodies. Lanes 1–4 (Rab7 mutants) and lanes 5–8 (Control). Original uncropped western blot image can be found in S1 Raw Images. (E) Fraction of cytokinetic defects in Rab7 null mutant pupal gonads (24–48 h post pupation), 11 independent experiments, and round spermatid count of about ($n = 1,100$). (F) Comparison of total cytokinetic defects between Rab7, Rab35 mutants, and Rab11DN (sum of all forms of cytokinetic defects including nuclear to mitochondria ratios 2:1, 3:1, 4:1, and irregularly sized nucleus). Each dot in each graph represents an independent experiment (n) that consisted of 10 individual testis/gonads. The data underlying the graphs shown in this figure can be found in S2 Data.
(PDF)

**S9 Fig. Endosomal volume measurement for various Rab mutant spermatocytes during cytokinesis.** Spermatocytes from all backgrounds were treated with PlyA2-mCherry and

imaged live to capture cytokinetic step. Endosome surfaces in 3D images were traced using surfaces tool in Imaris software by setting up appropriate thresholds. Images B, E, H, and K show the total endosomal surfaces traced using surfaces. Note that the plasma membrane and other areas of the images were not considered for calculating endosomal volume. Images C, F, I, and L show furrow-specific endosomal volume where white rectangle box indicates area chosen for measuring furrow-specific volume. (A–C) w[1118] spermatocytes, (D–F) Rab11 DN expressing spermatocytes, (G–I) Rab35 mutant spermatocytes, (J–L) Rab7 mutant spermatocytes. (M) Fraction of endosomes enriched at the furrow was calculated for all the backgrounds by dividing furrow-specific endosomal volume with total endosomal volume. Each dot in the graph represents an independent spermatocyte undergoing division from 3 independent experiments. w[1118] ($n = 15$), rab7 mutant ($n = 13$), Rab11 DN ($n = 16$), and rab35 mutant ($n = 15$). Statistics were performed using ordinary 1-way ANOVA Turkey's multiple comparisons in Prism9, ns = $p > 0.05$ and **** = $p < 0.0001$. The data underlying the graphs shown in this figure can be found in S2 Data.
(PDF)

**S10 Fig. Characterization of role of Rab35 in spermatogenesis.** (A) Confocal image showing germ cells in rab 35 mutant testis that are immunostained with Vasa antibody (green) and DNA (blue). (B) Phase contrast image showing round spermatids in testis squash preparations of rab35 mutant. (C) Confocal image showing elongated spermatids of rab35 mutant testis immunostained with cleaved caspase (DCP1) (red) and DNA (blue). (D) Confocal image showing early elongated spermatids in rab35 mutant testis squash preparations immunostained with alpha tubulin antibody (red) and DNA (blue). (E) Confocal images showing spermatocyte cysts that are expressing EYFP-Rab35 (endogenous promoter). Testis squash preparations of EYFP-Rab35 were immunostained with GFP antibody (green), ATP5A antibody to detect mitochondria (red), and DNA (blue). Cyst cell and its membranes are shown with arrow. (F) Confocal image showing early elongating spermatids in samples from E, cyst cells were shown with arrow and endosomes with arrow heads.
(PDF)

**S11 Fig. Sample processing for FIB-SEM.** (A) *Drosophila* cysts were dissected onto a gridded cover slip, after fixation and embedded in 4% agarose. The cyst of interest is circled and is the same as the cyst shown in Fig 7A. (B) Higher magnification image of boxed area in (A). (C) Light microscopy overview of full field before EM processing. (D) Same field after EM processing. The same pattern of dissected cysts is visible, suggesting spatial retention of sample. Trimmed area approximated with a dashed rectangle. (E) Field of view (orthogonal to Fig 7A) with doubly labeled cleavage furrow of interest circled. (F) Sectioning at the microtome proceeded until furrow of interest was almost exposed (red circle). FIB-SEM imaging was executed in the volume indicated by rectangle, in direction of arrow.
(PDF)

**S1 Movie. Live cell imaging of neuroblast asymmetric divisions in third instar larval brain.** GFP-tagged PH domain of PI(3,4,5)P$_3$ binding protein Step was expressed under the control of alpha tubulin84B promoter. Arrows indicate enrichment of GFP-Step on cleavage furrow membranes during cytokinesis.
(MP4)

**S2 Movie. Live cell imaging of an intact control *Drosophila* spermatocyte cyst undergoing meiosis I cytokinesis.** Spermatocytes are expressing myosin II regulatory light chain (RLC) tagged to GFP under its own promoter and Ubi-p63E promotor drives the expression of Feo-mCherry. An intact cyst is shown. Sqh-GFP-RLC localizes to actomyosin contractile ring. Feo-

mCherry localizes to central spindle.
(MP4)

**S3 Movie. Live cell imaging of an intact *cpes* mutant spermatocyte cyst undergoing meiosis I cytokinesis.** Mutant spermatocytes are expressing myosin II regulatory light chain (RLC) tagged to GFP under its own promoter and Feo-mCherry was expressed under the control of Ubi-p63E promotor. Sqh-GFP-RLC specifically localizes to actomyosin contractile ring. Feo-mCherry localizes to central spindle.
(MP4)

**S4 Movie. Live cell imaging of an intact CPES rescue spermatocyte cyst undergoing meiosis I cytokinesis.** The UAS-CPES is expressed using Tub-Gal4. Spermatocytes are also expressing Sqh-GFP-RLC and Feo-mCherry. Sqh-GFP-RLC specifically localizes to actomyosin contractile ring. Feo-mCherry localizes to central spindle.
(MP4)

**S5 Movie. Live cell imaging of an intact control *Drosophila* spermatocyte cyst undergoing meiosis I cytokinesis.** The UAS PLCδ-PH-EGFP is expressed using Tub-Gal4 and Feo-mCherry is expressed under the control of Ubi-p63E promoter. PLCδ-PH-EGFP specifically binds to $PI(4,5)P_2$ at the plasma membrane. Feo-mCherry localizes to central spindle.
(MP4)

**S6 Movie. Live cell imaging of an intact *cpes* mutant spermatocyte cyst undergoing meiosis I cytokinesis.** Tub-Gal4 drives the expression of UAS PLCδ-PH-EGFP and Ubi-p63E promoter drives the expression of Feo-mCherry. PLCδ-PH-EGFP specifically binds to $PI(4,5)P_2$ at the plasma membrane. Feo-mCherry localizes to central spindle.
(MP4)

**S7 Movie. Live cell imaging of an intact CPES rescue spermatocyte cyst undergoing meiosis I cytokinesis.** The UAS-CPES and UAS PLCδ-PH-EGFP are expressed using Tub-Gal4. The Feo-mCherry is expressed under the control of Ubi-p63E promoter. PLCδ-PH-EGFP specifically binds to $PI(4,5)P_2$ at the plasma membrane. Feo-mCherry localizes to central spindle.
(MP4)

**S8 Movie. Live cell imaging of an intact control *Drosophila* spermatocyte cyst undergoing meiosis I cytokinesis.** Alpha tubulin84B promoter drives the expression of GFP-tagged PH domain of Step that specifically binds to $PI(3,4,5)P_3$ at the plasma membrane. Feo-mCherry localizes to central spindle.
(MP4)

**S9 Movie. Live cell imaging of an intact *cpes* mutant spermatocyte cyst undergoing meiosis I cytokinesis.** The GFP-tagged PH domain of Step is expressed under the control of alpha tubulin84B promoter and Feo-mCherry is expressed under the control of Ubi-p63E promoter. The GFP-PH-Step specifically localizes to $PI(3,4,5)P_3$ at the plasma membrane and Feo-mCherry localizes to central spindle.
(MP4)

**S10 Movie. Live cell imaging of an intact CPES rescue spermatocyte cyst undergoing meiosis I cytokinesis.** PH domain of Step is tagged to GFP and expressed under the control of alpha tubulin84B promoter and Feo-mCherry is expressed under the control of Ubi-p63E promoter. The UAS-CPES expression is driven by Tub-Gal4. The GFP-PH-Step specifically localizes to $PI(3,4,5)P_3$ at the plasma membrane and Feo-mCherry localizes to central spindle.
(MP4)

**S11 Movie. Live cell imaging of EYFP-Rab7 expressing spermatocytes treated with recombinant CPE-binding protein tagged to mCherry (PlyA2-mCherry).** The EYFP-Rab7 is expressed under its own promotor.
(MP4)

**S12 Movie. Live cell imaging of EYFP-Rab11 expressing spermatocytes treated with recombinant PlyA2-mCherry.** EYFP-Rab11 is expressed under its own promotor.
(MP4)

**S13 Movie. Live cell imaging of wild-type spermatocytes treated with PlyA2-mCerry.**
(MP4)

**S14 Movie. Live cell imaging of EYFP Rab11 dominant negative (S25N) expressing spermatocytes.** UASP-EYFP-Rab11 (S25N) expression is regulated by Tubulin Gal80ts and Tubulin Gal4. Expression of UASP-EYFP-Rab11 (S25N) is induced by shifting third instar larvae from lower temperature 22˚C to higher temperature 30˚C to inactivate Gal80$^{ts}$. Four days post temperature shift testes were dissected, spermatocytes cysts were isolated and treated with PlyA2-mCherry before imaging.
(MP4)

**S15 Movie. Live cell imaging of Rab 35 mutant spermatocytes treated with PlyA2-mCherry.**
(MP4)

**S16 Movie. Live cell imaging of *rab7* mutant spermatocytes treated with recombinant CPE-binding protein tagged to mCherry (PlyA2-mCherry).**
(MP4)

**S17 Movie. FIB-SEM image stack corresponding to "Fig 7B".**
(MP4)

**S18 Movie. Live cell imaging of control *Drosophila* spermatocyte cyst undergoing meiosis I cytokinesis.** The EYFP-Rab7 is expressed under its own promotor. Tubulin Gal4 drives the expression mRFP-Anillin that localizes to contractile ring.
(MP4)

**S19 Movie. Movie showing the FIB-SEM volume reconstruction of the dividing cells in "Fig 7K," highlighting the points at which vesicle docking at the furrow could be captured.** Cells were 3D rendered in grayscale and overlaid by manually segmented 3D models of the cleavage furrow (red), undocked vesicles (green), and docked vesicles (blue).
(MP4)

**S20 Movie. Movie showing the FIB-SEM sub-volume reconstruction of the dividing cells in "Fig 7N," highlighting the points at which vesicle docking at the furrow could be captured.**
(MP4)

**S21 Movie. Live cell imaging of *cpes* mutant spermatocytes expressing EYFP-Rab7 under its own promotor and mRFP-Anillin driven by tubulin-Gal4.**
(MP4)

**S1 Raw Images. Unprocessed images for the TLC and blot in the paper.** Represented blot in the figure was shown in red box.
(PDF)

**S1 Data. Numerical data for Figs 1B–1D, 1F–1H, 2N–2P, and S1A–S1D.** Each tab in the file corresponds to numerical data used to produce respective figure panels. First tab includes the data for all individual replicates used in the study. Second tab shows mean and standard deviation derived from first tab.
(XLSX)

**S2 Data. Numerical data for Figs 2M, 3N, 3O, 4B–4D, 4F, 4H, 5Q, 6G, 6Q, S3A–S3C, S5D, S6A, S6B, S6D, S6E–S6H, S8A–S8C, S8E, S8F and S9M.**
(XLSX)

**S3 Data. Numerical data for S5H–S5K Fig.** First tab includes the data for all individual replicates used in this study. Second tab shows mean and standard deviation derived from first tab. Each other tab in the file corresponds to numerical data used to produce respective figure panels.
(XLSX)

**S1 Table. Description of primers used in the study.**
(DOCX)

# Acknowledgments

We thank Mr. Aayush Bhatwadekar for help with movies and rendering for the FIB-SEM images.

# Author Contributions

**Conceptualization:** Govind Kunduri, Usha Acharya, Jairaj K. Acharya.

**Data curation:** Govind Kunduri, Si-Hung Le, Valentina Baena, Nagampalli Vijaykrishna, Adam Harned, Kunio Nagashima, Kedar Narayan, Takeshi Bamba.

**Formal analysis:** Govind Kunduri, Si-Hung Le, Valentina Baena, Nagampalli Vijaykrishna, Izumi Yoshihiro, Kedar Narayan, Takeshi Bamba, Usha Acharya, Jairaj K. Acharya.

**Funding acquisition:** Kedar Narayan, Takeshi Bamba, Jairaj K. Acharya.

**Investigation:** Govind Kunduri, Si-Hung Le, Valentina Baena, Nagampalli Vijaykrishna, Kedar Narayan, Usha Acharya, Jairaj K. Acharya.

**Methodology:** Govind Kunduri, Si-Hung Le, Valentina Baena, Nagampalli Vijaykrishna, Adam Harned, Kunio Nagashima, Kedar Narayan, Takeshi Bamba.

**Project administration:** Daniel Blankenberg, Izumi Yoshihiro, Kedar Narayan, Takeshi Bamba, Usha Acharya, Jairaj K. Acharya.

**Resources:** Kedar Narayan, Takeshi Bamba, Jairaj K. Acharya.

**Software:** Valentina Baena, Nagampalli Vijaykrishna, Takeshi Bamba.

**Supervision:** Daniel Blankenberg, Izumi Yoshihiro, Kedar Narayan, Takeshi Bamba, Usha Acharya, Jairaj K. Acharya.

**Validation:** Govind Kunduri, Si-Hung Le, Valentina Baena, Nagampalli Vijaykrishna, Takeshi Bamba, Usha Acharya, Jairaj K. Acharya.

**Visualization:** Govind Kunduri, Si-Hung Le, Valentina Baena, Nagampalli Vijaykrishna, Kunio Nagashima, Usha Acharya, Jairaj K. Acharya.

**Writing – original draft:** Govind Kunduri, Usha Acharya, Jairaj K. Acharya.

**Writing – review & editing:** Govind Kunduri, Valentina Baena, Daniel Blankenberg, Izumi Yoshihiro, Kedar Narayan, Takeshi Bamba, Usha Acharya, Jairaj K. Acharya.

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
