## [Editor Report · Decision Letter 0]

10 Mar 2022

Dear Dr Acharya, 

Thank you for submitting your revised manuscript entitled "Endosomes deliver ceramide phosphoethanolamine with unique acyl chain anchors to the cleavage furrow during male meiotic cytokinesis" for consideration as a Research Article by PLOS Biology. Please accept my apologies for the delay in getting back to you. 

Your revision and rebuttal has now been evaluated by the PLOS Biology editorial staff, as well as by the academic editor who handled your previous submission, and I am writing to let you know that we would like to send your submission back out for review by the original reviewers. 

However, before we can send your manuscript to the reviewers, we need you to complete your submission by providing the metadata that is required for full assessment. To this end, please login to Editorial Manager where you will find the paper in the 'Submissions Needing Revisions' folder on your homepage. Please click 'Revise Submission' from the Action Links and complete all additional questions in the submission questionnaire.

Once your full submission is complete, your paper will undergo a series of checks in preparation for peer review. Once your manuscript has passed the checks it will be sent out for review. To provide the metadata for your submission, please Login to Editorial Manager (https://www.editorialmanager.com/pbiology) within two working days, i.e. by Mar 12 2022 11:59PM.

Given the disruptions resulting from the ongoing COVID-19 pandemic, please expect some delays in the editorial process. We apologise in advance for any inconvenience caused and will do our best to minimize impact as far as possible.

Kind regards,

Richard

Richard Hodge, PhD

Associate Editor, PLOS Biology

rhodge@plos.org

PLOS

---

## [Decision Letter · Decision Letter 1]

5 Apr 2022

Dear Dr Acharya,

Thank you for submitting your revised manuscript "Endosomes deliver ceramide phosphoethanolamine with unique acyl chain anchors to the cleavage furrow during male meiotic cytokinesis" as a Research Article for review by PLOS Biology. As with all papers reviewed by the journal, yours was assessed and discussed by the PLOS Biology editors, by an academic editor with relevant expertise, and by three independent reviewers. Please note that whilst your manuscript was sent back to the original reviewers (R1-R4), we were only able to successfully recruit Reviewers #2 and #3 to provide a review. As a result, we recruited an additional reviewer (R5) to review the revised version of the manuscript.

As you can see, the original reviewers are satisfied and think that most of their concerns have been addressed. Whilst Reviewer #5 finds your study to be interesting and well-done, s/he suggests several experiments to further support the findings that CPE is delivered to the furrow by endosomes and is important for cytokinesis, as well as providing additional analyses and quantifications for the cytokinetic defects and PI(4,5)P2 distribution data. After discussions with the Academic Editor and the rest of the editorial team, we think that the suggested experiments would strengthen the paper. Nevertheless, since the second part of comments #2 and #3 may represent a lot of work, we will not make these specific experiments a requirement for publication.

In light of the reviews, we will not be able to accept the current version of the manuscript, but we would welcome re-submission of a much-revised version that takes into account the reviewers' comments. We cannot make any decision about publication until we have seen the revised manuscript and your response to the reviewers' comments. Your revised manuscript is also likely to be sent for further evaluation by the reviewers.

We expect to receive your revised manuscript within 3 months. Please email us (plosbiology@plos.org) if you have any questions or concerns, or would like to request an extension. At this stage, your manuscript remains formally under active consideration at our journal; please notify us by email if you do not intend to submit a revision so that we may end consideration of the manuscript at PLOS Biology.

**IMPORTANT - SUBMITTING YOUR REVISION**

*Re-submission Checklist*

*Published Peer Review*

*PLOS Data Policy*

*Blot and Gel Data Policy*

Sincerely,

Richard

Richard Hodge, PhD

Associate Editor, PLOS Biology

rhodge@plos.org

REVIEWS:

Reviewer #2: Kunduri and co-workers identify anew role for a lipid in male meiotic cytokinesis. They take a a series rigourous and elegant approaches to interrogate not only the identity of the lipid, but where in the cytokinetic pathway it functions and implicate a novel pathway to deliver the lipid to the cytokinetic machinery. This comprehensive study will be a great interest to the general field of cell division.

Reviewer #3: The revised version of the manuscript by Kunduri and colleagues is greatly improved. The authors have answered to all my major concerns. The novel organization of the results and figures is clearer and more convincing. The description of Rab phenotypes during cytokinesis and CLEM data are very important additions to the manuscript. The role of MVBs in delivering membranes to the cleavage furrow is an interesting hypothesis, but it is not demonstrated in this manuscript.

Reviewer #5: This is first time that I read this manuscript, and I understand that this is a revision. Overall, this is a very interesting paper that, in my opinion, should be published in Plos Biology. In particular, the genetic, lipidomic and EM data are of excellent quality, and represent a huge amount of work. However, I recommend that the following points are addressed before the paper is published. 

1- A more detailed description of the cytokinetic defects is needed. Based on the fluorescent Feo+sqh movies, the authors should provide the rate and magnitude of furrow ingression for individual cells in control vs mutant situations. From movie S3, it seems that the actin ring detaches from the plasma membrane in most instances, which is not well captured in fixed samples (Fig. 4B). In addition, Feo is present on linear structures in early meiosis in the cpes mutant (movie S3) but not in controls (movie S4): how are the spindle and the microtubules in the cpes mutant? A basic description of the chromosomes is also needed: do chromosomes struggle to align properly (I noticed BUBR1 in Fig. S6E)?

2- The impact of the paper would be further increased if the authors could demonstrate that i) CPE is delivered by endosomes to the furrow membrane and ii) this is important for cytokinesis. In the present manuscript, there are only interesting correlative evidence. In this respect, the conclusion (line 639) is an overstatement.

First, is CPE locally enriched at the furrow MEMBRANE when Rab7 endosomes are in its vicinity in control cells? If Rab7 endosomes deliver CPE to the furrow for proper ingression, this would be expected. 

Second, the authors should try to find a mutant situation where the CPE endosomes are not properly localized to the furrow. Although Rab7 mutants failed to show this, what does happen in the Rab11 DN situation (this is only touched in the discussion)? Based on movies, how does the CPE probe (especially at the PM of the furrow) exactly behave during cytokinesis in the Rab7 mutant or in the Rab11 DN mutant (beside single movies, quantifications are needed)? 

I also suggest that the authors should have a look to Rab35 mutants, to uncouple CPE endosomes from the furrow (YFP-Rab35 WT and DN flies are available). Indeed, i) Rab35 is a recycling endosome marker and is required for cytokinesis (many papers since 2006), ii) Rab35 controls MVB fusion to the PM (PMID 20404108), iii) Rab35 is a negative regulator of mTOR during myelin growth (PMID 32503983). This might be an interesting connection since TOR is the first hit in Fig. S6C and CPE mimics GalCer, which is a major sphingolipid in myelin sheath (as discussed, line 814).

3- The absence of defects in PI(4,5)P2 distribution in cpes mutants should be quantified and analyzed in more detail. 

First, it seems that PI(4,5)P2 distribution is not uniform in controls (less present at the furrow, compared to the pole, Fig. 4C) whereas PI(4,5)P2 is evenly distributed along the PM in cpes mutant. 

Second, there is a striking increase of PI(4,5)P2 amounts (based on the PH-GFP probe) at the PM in mutants (Fig. 4C) with more membrane ruffles (movie S6). Is this true? In this case, would reducing PI(4,5)P2 (e.g. by reducing PI4P-5-kinase activity in an heterozygous Skittle mutant) rescue the cpes mutant phenotype?

Minor points

1- Fig 4B: please provide quantifications in controls.

2- The picture in Fig. 5P does not reflect the quantification for Rab11/CPE colocalization in Fig 5Q. Either the picture is not representative, or the quantification is not accurate.

3- Fig. 6: please provide movies and quantification of PlyA2 in cells that do no express Rab proteins (see major point #2)

4- Fig. 8E: the fusion of a small endosome to an MVB is misleading. Endosomes first fuse together to form large early endosomes that eventually mature into MVBs with ILVs.

5- References should be improved. Some cited reviews are good but outdated, original papers are not always cited or previous published results are not associated with references. For instance, a paper that seems important to cite and discuss in the light of the present study is PMID 18804373 (line 84).

6- Line 310: to validate the conclusion, it would be important to provide a quantification of the ceramid levels (as in Fig. 2O) upon ceramidase expression.

7- Change PIP2 and PIP3 to PI(4,5)P2 and PI(3,4,5)P3, respectively, for clarity.

8- Line 642: missing BDSC#

---

## [Editor Report · Decision Letter 2]

11 Jul 2022

Dear Dr Acharya,

Thank you for submitting your revised manuscript "Endosomes deliver ceramide phosphoethanolamine with unique acyl chain anchors to the cleavage furrow during male meiotic cytokinesis" for publication as a Research Article at PLOS Biology. This revised version of your manuscript has been evaluated by the PLOS Biology editors and the Academic Editor.

Based on our Academic Editor's assessment of your revision, I am pleased to say that we are likely to accept this manuscript for publication, provided you satisfactorily address the following data and other policy-related requests that I have provided below (A-F):

(A) We would like to propose the following modification to the title, in order to make it more compelling for our broad readership:

‘Delivery of ceramide phosphoethanolamine lipids to the cleavage furrow through the endocytic pathway is essential for male meiotic cytokinesis’

(B) You may be aware of the PLOS Data Policy, which requires that all data be made available without restriction: http://journals.plos.org/plosbiology/s/data-availability. For more information, please also see this editorial: http://dx.doi.org/10.1371/journal.pbio.1001797

- Supplementary files (e.g., excel). Please ensure that all data files are uploaded as 'Supporting Information' and are invariably referred to (in the manuscript, figure legends, and the Description field when uploading your files) using the following format verbatim: S1 Data, S2 Data, etc. Multiple panels of a single or even several figures can be included as multiple sheets in one excel file that is saved using exactly the following convention: S1_Data.xlsx (using an underscore).

- Deposition in a publicly available repository. Please also provide the accession code or a reviewer link so that we may view your data before publication.

Figure 1B-D, 1F-H, 2M-P, 3N-O, 4B-D, 4F, 4H, 5Q, 6G, 6Q, S1A-D, S3A-C, S5D, S5H-K, S6A-H, S8A-C, S8E-F, S9M

(C) We suggest that you deposit the RNA-sequencing data in a publicly available repository, such as the GEO. Please ensure that you provide the accession number/URL of the deposition in the Data statement in the online submission form. 

(D) Please also ensure that figure legends in your manuscript include information on *WHERE THE UNDERLYING DATA CAN BE FOUND*, and ensure your supplemental data file/s has a legend.

(E) We require the original, uncropped and minimally adjusted images supporting all blot and gel results reported in the following figures:

Figure S5L, S8D

We will require these files before a manuscript can be accepted so please prepare and upload them now. Please carefully read our guidelines for how to prepare and upload this data: https://journals.plos.org/plosbiology/s/figures#loc-blot-and-gel-reporting-requirements.

(F) Please note that per journal policy, the model system/species studied should be clearly stated in the abstract of your manuscript.

We expect to receive your revised manuscript within two weeks. 

*Published Peer Review History*

*Press*

Sincerely,

Richard

Richard Hodge, PhD

Associate Editor, PLOS Biology

rhodge@plos.org

PLOS

---

## [Editor Report · Decision Letter 3]

2 Aug 2022

Dear Dr Acharya,

On behalf of my colleagues and the Academic Editor, Mariana Wolfner, I am pleased to say that we can accept your manuscript for publication, provided you address any remaining formatting and reporting issues. These will be detailed in an email you should receive within 2-3 business days from our colleagues in the journal operations team; no action is required from you until then. Please note that we will not be able to formally accept your manuscript and schedule it for publication until you have completed any requested changes.

PRESS

Sincerely, 

Richard

Richard Hodge, PhD

Associate Editor, PLOS Biology

rhodge@plos.org

PLOS
